# INFORMATION THEORETIC META LEARNING WITH GAUSSIAN PROCESSES

## ABSTRACT

We formulate meta learning using information theoretic concepts such as mutual information and the information bottleneck. The idea is to learn a stochastic representation or encoding of the task description, given by a training or support set, that is highly informative about predicting the validation set. By making use of variational approximations to the mutual information, we derive a general and tractable framework for meta learning. We particularly develop new memory-based meta learning algorithms based on Gaussian processes and derive extensions that combine memory and gradient-based meta learning. We demonstrate our method on few-shot regression and classification by using standard benchmarks such as Omniglot, mini-Imagenet and Augmented Omniglot.

## 1 INTRODUCTION

Meta learning (Ravi & Larochelle, 2017; Vinyals et al., 2016; Edwards & Storkey, 2017; Finn et al., 2017; Lacoste et al., 2019; Nichol et al., 2018) and few-shot learning (Li et al., 2006; Lake et al., 2011) aim to derive data efficient learning algorithms that can rapidly adapt to new tasks. Such systems require training deep neural networks from a set of tasks drawn from a common distribution, where each task is described by a small amount of experience, typically divided into a training or support set and a validation set. By sharing information across tasks the neural network can learn to rapidly adapt to new tasks and generalize from few examples at test time.

Several few-shot learning algorithms use memory-based (Vinyals et al., 2016; Ravi & Larochelle, 2017) or gradient-based procedures (Finn et al., 2017; Nichol et al., 2018), with the gradient-based model agnostic meta learning algorithm (MAML) by Finn et al. (2017) being very influential in the literature. Despite the success of specific schemes, one fundamental issue in meta learning is concerned with deriving unified principles that can allow to relate different approaches and invent new schemes. While there exist probabilistic interpretations of existing methods, such as the approximate Bayesian inference approach (Grant et al., 2018; Finn et al., 2018; Yoon et al., 2018) and the related conditional probability modelling approach (Garnelo et al., 2018; Gordon et al., 2019), meta learning still lacks of a general and tractable learning principle that can help to get a better understanding of existing algorithms and derive new methods.

To this end, the main contribution of this paper is to introduce an information theoretic view of meta learning, by utilizing tools such as the mutual information and the information bottleneck (Cover & Thomas, 2006; Tishby et al., 1999). Given that each task consists of a support or training set and a target or validation set, we consider the information bottleneck principle, introduced by Tishby et al. (1999), which can learn a stochastic encoding of the support set that is highly informative about predicting the validation set. Such stochastic encoding is optimized through the difference between two mutual informations, so that the encoding compresses the training set into a representation that can predict well the validation set. By exploiting recent variational approximations to the information bottleneck (Alemi et al., 2017; Chalk et al., 2016; Achille & Soatto, 2016) that make use of variational lower bounds on the mutual information (Barber & Agakov, 2003), we derive a general and tractable framework for meta learning. Such framework can allow us to re-interpret gradient-based algorithms, such as MAML, and also derive new methods.

Based on the variational information bottleneck (VIB) framework (Alemi et al., 2017; Chalk et al., 2016; Achille & Soatto, 2016), we introduce a new memory-based algorithm for supervised few-shot learning (right panel in Figure 1) based on Gaussian processes (Rasmussen & Williams, 2006)

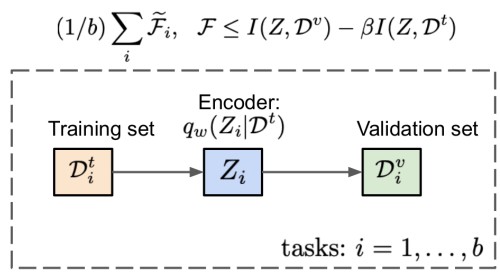 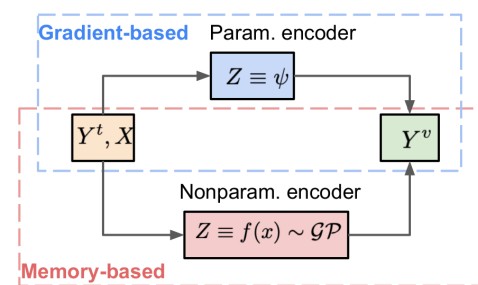

**Figure 1:** (**left**) Meta learning with the information bottleneck. $Z_i \sim q_w(Z_i|\mathcal{D}_i^t)$ is the encoder we wish to optimize to compress the task training set $\mathcal{D}_i^t$ (minimize the mutual information $I(Z_i, \mathcal{D}_i^t)$) and predict well the validation set $\mathcal{D}_i^v$ (maximize $I(Z_i, \mathcal{D}_i^v)$); see Section 2. (**right**) Specialization to supervised few-shot learning where for each task we have input-output data. Gradient-based algorithms, such as MAML, and Gaussian process memory-based methods (proposed in this paper) are instances of the framework; see Section 2.2 and 3.

and deep neural kernels (Wilson et al., 2016) that offers a kernel-based Bayesian view of a memory system. With Gaussian processes, the underlying encoding takes the form of a *non-parametric* function that follows a stochastic process amortized by the training set. Further, we show that VIB gives rise to gradient-based meta learning methods, such as MAML, when combined with *parametric* encodings corresponding to model parameters or weights, and based on this we derive a stochastic MAML algorithm. In an additional scheme, we show that our framework can naturally allow for combinations of memory and gradient-based meta learning by constructing suitable encodings, and we derive such an algorithm that combines Gaussian processes with MAML. We demonstrate our methods on few-shot regression and classification by using standard benchmarks such as Omniglot, mini-Imagenet and Augmented Omniglot.

## 2 META LEARNING WITH INFORMATION BOTTLENECK

Suppose we wish to learn from a distribution of tasks. During training for each task we observe a pair consisted of a *task description* represented by the support or training set $\mathcal{D}^t$ and *task validation* represented by the target or validation set $\mathcal{D}^v$. At test time only $\mathcal{D}^t$ will be given and the learning algorithm should rapidly adapt to form predictions on $\mathcal{D}^v$ or on further test data.

We wish to formulate meta learning using information theoretic concepts such as mutual information and the information bottleneck (Tishby et al., 1999). The idea is to learn a stochastic representation or encoding of the task description $\mathcal{D}^t$ that is highly informative about predicting $\mathcal{D}^v$. We introduce a random variable, $Z$, associated with this encoding drawn from a distribution $q_w(Z|\mathcal{D}^t)$ parametrized by $w$. Given this encoding the full joint distribution is written as

$$q_w(\mathcal{D}^v, \mathcal{D}^t, Z) = q_w(Z|\mathcal{D}^t)p(\mathcal{D}^v, \mathcal{D}^t), \qquad (1)$$

where $p(\mathcal{D}^v, \mathcal{D}^t)$ denotes the unknown data distribution over $\mathcal{D}^t$ and $\mathcal{D}^v$. In equation 1 and throughout the paper we use the convention that the full joint as well as any marginal or conditional that depends on $Z$ is denoted by $q_w$ (emphasizing the dependence on the parametrized encoder), while corresponding quantities over data $\mathcal{D}^t, \mathcal{D}^v$ are denoted by $p$. Eg. from the above we can express a $Z$-dependent marginal such as, $q_w(Z, D^v) = \int q_w(Z|\mathcal{D}^t)p(\mathcal{D}^v, \mathcal{D}^t)d\mathcal{D}^t$.

To tune $w$ we would like to maximize the mutual information between $Z$ and the target set $\mathcal{D}^v$, denoted by $I(Z, \mathcal{D}^v)$. A trivial way to obtain a maximally informative representation is to set $Z = \mathcal{D}^t$, which does not provide a useful representation. Thus, the information bottleneck (IB) principle (Tishby et al., 1999) adds a model complexity penalty to the maximization of $I(Z, \mathcal{D}^v)$ which promotes an encoding $Z$ that is highly compressive of $\mathcal{D}^t$, i.e. for which $I(Z, \mathcal{D}^t)$ is minimized. This leads to the IB objective:

$$\mathcal{L}_{IB}(w) = I(Z, \mathcal{D}^v) - \beta I(Z, \mathcal{D}^t), \qquad (2)$$

where $\beta \geq 0$ is a hyperparameter. Nevertheless, in order to use IB for meta learning we need to approximate the mutual information terms $I(Z, \mathcal{D}^v)$ and $I(Z, \mathcal{D}^t)$, which are both intractable since they depend on the unknown data distribution $p(\mathcal{D}^v, \mathcal{D}^t)$. To overcome this, we will consider variational approximations by following similar arguments to the variational IB approach (Alemi

et al., 2017) that was introduced for supervised learning of a single task, which allows us to express a tractable lower bound on $\mathcal{L}_{IB}(w)$ by lower bounding $I(Z, \mathcal{D}^v)$ and upper bounding $I(Z, \mathcal{D}^t)$.

## 2.1 VARIATIONAL INFORMATION BOTTLENECK (VIB) FOR META LEARNING

To construct a bound $\mathcal{F} \leq \mathcal{L}_{IB}(w)$ we need first to lower bound $I(Z, \mathcal{D}^v)$, which is written as

$$I(Z, \mathcal{D}^v) = \mathrm{KL}\left[q_w(Z, \mathcal{D}^v)||q_w(Z)p(\mathcal{D}^v)\right] = \mathbb{E}_{q_w(Z, \mathcal{D}^v)}\left[\log \frac{q_w(\mathcal{D}^v|Z)}{p(\mathcal{D}^v)}\right], \tag{3}$$

where KL denotes the Kullback-Leibler divergence and $q_w(\mathcal{D}^v|Z) = \frac{\int q_w(Z|\mathcal{D}^t)p(\mathcal{D}^v, \mathcal{D}^t)d\mathcal{D}^t}{\int q_w(Z|\mathcal{D}^t)p(\mathcal{D}^v, \mathcal{D}^t)d\mathcal{D}^t d\mathcal{D}^v}$ is intractable since we do not know the analytic form of the data distribution $p(\mathcal{D}^v, \mathcal{D}^t)$. To lower bound $I(Z, \mathcal{D}^v)$, we follow Barber & Agakov (2003) (see Appendix A.1) by introducing a *decoder* model $p_\theta(\mathcal{D}^v|Z)$ to approximate the intractable $q_w(\mathcal{D}^v|Z)$ where $\theta$ are additional parameters[1],

$$I(Z, \mathcal{D}^v) \geq \mathbb{E}_{q_w(Z, \mathcal{D}^v)}\left[\log \frac{p_\theta(\mathcal{D}^v|Z)}{p(\mathcal{D}^v)}\right] = \mathbb{E}_{q_w(Z, \mathcal{D}^v)}\left[\log p_\theta(\mathcal{D}^v|Z)\right] + \mathcal{H}(\mathcal{D}^v), \tag{4}$$

where the entropy $\mathcal{H}(\mathcal{D}^v)$ is just a constant that does not depend on the tunable parameters $(\theta, w)$. Furthermore, to deal with the second intractable mutual information $I(Z, \mathcal{D}^t)$ so that to maintain an lower bound on $\mathcal{L}_{IB}(w)$ we need to upper bound this term. Note that

$$I(Z, \mathcal{D}^t) = \mathbb{E}_{q_w(Z, \mathcal{D}^t)}\left[\log \frac{q_w(Z|\mathcal{D}^t)}{q_w(Z)}\right],$$

where $q_w(Z) = \int q_w(Z|\mathcal{D}^t)p(\mathcal{D}^t)d\mathcal{D}^t$ is intractable since, e.g. it involves the unknown data distribution $p(\mathcal{D}^t)$. By working similarly as before, we can approximate $q_w(Z)$ by a tractable *prior* model distribution $p_\theta(Z)$ which leads to the following upper bound on the mutual information,

$$I(Z, \mathcal{D}^t) \leq \mathbb{E}_{q_w(Z, \mathcal{D}^t)}\left[\log \frac{q_w(Z|\mathcal{D}^t)}{p_\theta(Z)}\right]. \tag{5}$$

Then, by combining the two bounds we obtain the overall bound, $\mathcal{F}(\theta, w) + \mathcal{H}(\mathcal{D}^v) \leq \mathcal{L}_{IB}(w)$:

$$\mathcal{F}(\theta, w) = \mathbb{E}_{q_w(Z, \mathcal{D}^v)}\left[\log p_\theta(\mathcal{D}^v|Z)\right] - \beta\mathbb{E}_{q_w(Z, \mathcal{D}^t)}\left[\log \frac{q_w(Z|\mathcal{D}^t)}{p_\theta(Z)}\right],$$

where the constant $\mathcal{H}(\mathcal{D}^v)$ is dropped from the objective function. Given a set of task pairs $\{\mathcal{D}_i^t, \mathcal{D}_i^v\}_{i=1}^b$, where each $(\mathcal{D}_i^t, \mathcal{D}_i^v) \sim p(\mathcal{D}^v, \mathcal{D}^t)$, during meta-training the objective function for learning $(\theta, w)$ reduces to the maximization of the empirical average, $\frac{1}{b}\sum_i \widetilde{\mathcal{F}}_i(\theta, w)$, where each $\widetilde{\mathcal{F}}_i(\theta, w)$ is an unbiased estimate of $\mathcal{F}(\theta, w)$ (see Appendix A.2) and is given by

$$\widetilde{\mathcal{F}}_i(w, \theta) = \mathbb{E}_{q_w(Z_i|\mathcal{D}_i^t)}\left[\log p_\theta(\mathcal{D}_i^v|Z_i)\right] - \beta\mathrm{KL}[q_w(Z_i|\mathcal{D}_i^t)||p_\theta(Z_i)]. \tag{6}$$

The meta-training procedure is carried out in different episodes where at each step we receive a minibatch of task pairs and perform a stochastic gradient maximization step.

The objective in equation 6 is similar to variational inference objectives for meta learning (Ravi & Beatson, 2019). In particular, it can be viewed as an evidence lower bound (ELBO) on the validation set log marginal likelihood, $\log \int p_\theta(\mathcal{D}_i^v|Z_i)p_\theta(Z_i)dZ_i$, with the differences: (i) there is the hyperparameter $\beta$ in front of the KL term and (ii) the *variational distribution* $q_w(Z_i|\mathcal{D}_i^t)$ in equation 6 is more restricted than in standard variational inference, since $q_w(Z_i|\mathcal{D}_i^t)$ now acts as a *stochastic bottleneck* that encodes the support set $\mathcal{D}_i^t$ (i.e. it is amortized by $\mathcal{D}_i^t$) and via the term $\mathbb{E}_{q_w(Z_i|\mathcal{D}_i^t)}[\log p_\theta(\mathcal{D}_i^v|Z_i)]$ it is optimized to reconstruct the validation set.

## 2.2 INFORMATION THEORETIC VIEW OF MAML-TYPE METHODS

MAML (Finn et al., 2017) is a special case of our framework. To see this, suppose that the task encoding variable $Z_i$ for the $i$-th task coincides with a vector of some task-specific model parameters or neural network weights $\psi_i$, so that $p_\theta(\mathcal{D}_i^v|Z_i) \equiv p(\mathcal{D}_i^v|\psi_i)$ and $p_\theta(Z_i)$ reduces to a prior

---

[1]The lower bounds are valid even when the parameters $w$ of the encoder $q_w(Z|\mathcal{D}^t)$ and $\theta$ of the decoder $p_\theta(\mathcal{D}^v|Z)$ (and prior $p_\theta(Z)$) have shared components, e.g. are parameters of the same neural architecture.

$p_\theta(\psi_i)$ over these parameters. The MAML approach (Finn et al., 2017) tries to find a shared initial parameter value $\theta$ so that few gradient steps based on the support set objective, $\log p(\mathcal{D}_i^t|\theta)$, lead to a task-specific parameter value $\psi_i$ with good generalization on the validation set. MAML estimates the task parameters by $\psi_i = \theta + \Delta(\theta, \mathcal{D}_i^t)$, where $\Delta(\theta, \mathcal{D}_i^t)$ denotes the inner loop adaptation steps which for a single gradient step is just $\rho \nabla_\theta \log p(\mathcal{D}_i^t|\theta)$ and where $\rho$ is a step size. By setting $\beta = 0$ and by using a deterministic Dirac measure encoder, $\delta_{\psi_i, \theta + \Delta(\theta, \mathcal{D}_i^t)}$, the VIB objective from equation 6 reduces to the standard MAML objective, $\widetilde{\mathcal{F}}_i(\theta) = \log p(\mathcal{D}_i^v|\theta + \Delta(\theta, \mathcal{D}_i^t))$.

We can construct a generalization of MAML by making the encoder stochastic, e.g. $q_{\theta,s}(\psi_i|\mathcal{D}_i^t) = \mathcal{N}(\psi_i|\theta + \Delta(\theta, \mathcal{D}_i^t), sI)$ where $s$ is a scalar variance parameter. Then, the objective becomes

$$\widetilde{\mathcal{F}}_i(\theta, s) = \mathbb{E}_{\mathcal{N}(\epsilon|0,I)}\left[\log p(\mathcal{D}^v|\theta + \Delta(\theta, \mathcal{D}_i^t) + \sqrt{s}\epsilon)\right] - \beta\mathrm{KL}\left[q_{\theta,s}(\psi_i|\mathcal{D}_i^t)||p_\theta(\psi_i)\right],$$

where we reparametrized the expectation suggesting the use of the reparametrization trick (Kingma & Welling, 2013) for stochastic optimization of the meta parameters $(\theta, s)$. In the experiments we will investigate an instance of the above where $p_\theta(\psi_i)$ follows the hierarchical Gaussian form $\mathcal{N}(\psi_i|\theta, sI)$, which views each task-specific parameter $\psi_i$ as a randomized version of $\theta$ and with $s$ being the same variance used by the encoder. For such prior the KL divergence term reduces to $-\frac{1}{2s}||\Delta(\theta, \mathcal{D}_i^t)||^2$, which penalizes large values of the inner adaptation steps and small values of $s$.

## 2.3 VIB FOR SUPERVISED META LEARNING

Here, we explain how to adapt the VIB principle to supervised meta learning. Suppose the learning problem involves few-shot supervised learning where for each task we wish to predict outputs or labels given corresponding inputs. Let us denote the task support set as $\mathcal{D}^t = (Y^t, X^t)$, where $Y^t = \{y_j^t\}_{j=1}^{n^t}$ and $X^t = \{x_j^t\}_{j=1}^{n^t}$ denote the output and input observations. Similarly, we write $\mathcal{D}^v = (Y^v, X^v)$ for the validation set. During meta-testing, for any novel task we observe the support set $\mathcal{D}_*^t = (Y_*^t, X_*^t)$ together with the test inputs $X_*^v$ and the goal is to predict the test outputs $Y_*^v$. This suggests that we can construct a task encoder distribution of the form $q_w(Z|Y^t, X)$ that depends on the training outputs $Y^t$ and generally on all inputs $X = (X^t, X^v)$.[2] We would like to train this encoder so that $Z$ becomes highly predictive about the validation outputs $Y^v$ and simultaneously compressive about $Y^t$. Then, a suitable VIB objective can be based on approximating the *input-conditioned* information bottleneck, $I(Z, Y^v|X) - \beta I(Z, Y^t|X)$ i.e. where both $I(Z, Y^v|X)$ and $I(Z, Y^t|X)$ are conditional mutual informations (see Appendix A.3). By following similar arguments as those in Section 2.1, we can lower bound this objective and finally approximate it by an unbiased empirical average, $\frac{1}{b}\sum_{i=1}^b \widetilde{\mathcal{F}}_i(\theta, w)$, where

$$\widetilde{\mathcal{F}}_i(\theta, w) = \mathbb{E}_{q_w(Z_i|Y_i^t, X_i)}\left[\log p_\theta(Y_i^v|Z_i, X_i)\right] - \beta\mathrm{KL}\left[q_w(Z_i|Y_i^t, X_i)||p_\theta(Z_i|X_i)\right], \quad (7)$$

and where $p_\theta(Y_i^v|X_i^v, Z_i)$ and $p_\theta(Z_i|X_i)$ are the decoder and prior model distributions introduced when applying the variational approximation. A detailed derivation can be found in Appendix A.3. Equation 7 provides the general form of the VIB objective suitable for supervised meta learning. The supervised version of MAML is expressed as a special case by following the arguments of Section 2.2. In Section 3, we particularize the above by combining it with a Gaussian process model.

## 3 SUPERVISED META LEARNING WITH GAUSSIAN PROCESSES

In this section we introduce VIB-based meta learning algorithms using Gaussian processes (GPs), which are suitable for few-shot supervised learning. In Section 3.1 we introduce a memory-based system, while in Section 3.2 we further combine it with gradient-based meta learning.

### 3.1 A GAUSSIAN PROCESS MEMORY-BASED METHOD

To use the VIB framework for few-shot supervised learning, as described in Section 2.3, for each $i$-th task we need to specify the encoding variable $Z_i$ together with the encoder $q_w(Z_i|Y_i^t, X_i)$, the decoder over the validation outputs $p_\theta(Y_i^v|Z_i, X_i)$ and the prior model $p_\theta(Z_i|X_i)$. Here, we construct these quantities by using a GP model (Rasmussen & Williams, 2006).

---

[2]Dependence on all inputs can allow to explain both transductive and non-transductive settings for meta learning (Bronskill et al., 2020; Finn et al., 2017; Nichol et al., 2018) as special cases; see Appendix B.

We denote the unknown task-specific function that solves the $i$-th task by $f_i(x)$ and we assume that a priori (before observing any task data) this function is a draw from a GP, i.e. $f_i(x) \sim \mathcal{GP}(0, k_\theta(x, x'))$, where $k_\theta$ denotes the kernel function. Without loss of generality we shall use a deep kernel function where $f_i(x)$ is a linear function of a deep neural network feature vector $\phi(x; \theta)$ with task-specific Gaussian weights, i.e. $f_i(x) = \phi(x; \theta)^\top \theta_i^{out}, \theta_i^{out} \sim \mathcal{N}(\theta_i^{out}|0, \sigma_f^2 I)$. Such function can be equivalently viewed as a GP sample: $f_i(x) \sim \mathcal{GP}(0, k_\theta(x, x'))$, $k_\theta(x, x') = \sigma_f^2 \phi(x; \theta)^\top \phi(x'; \theta)$. In this functional space view the task-specific output weights $\theta_i^{out}$ have been marginalized out and we are left with the feature vector parameters $\theta$ shared across tasks.[3]

Suppose now that we observe the task data, i.e. the support $\mathcal{D}_i^t = (Y_i^t, X_i^t)$ and validation $\mathcal{D}_i^v = (Y_i^v, X_i^v)$ sets, so that we can evaluate the task function on all task inputs $X_i = (X_i^t, X_i^v)$. Let $f_{i,j}^v \equiv f(x_{i,j}^v)$ denote the function value at the validation input $x_{i,j}^v$, associated with output $y_{i,j}^v$, while the vector of all such values is denoted by $\mathbf{f}_i^v = \{f_{i,j}^v\}_{j=1}^{n^v}$. Similarly, the vector of function values at the training inputs $X_i^t$ is $\mathbf{f}_i^t$. In the VIB framework, we specify the task encoding variable $Z_i$ to be the full set of function values, $Z_i \equiv (\mathbf{f}_i^v, \mathbf{f}_i^t)$, and we further choose the prior model for this encoding to be the GP prior, $p_\theta(Z_i|X_i) \equiv p(\mathbf{f}_i^v, \mathbf{f}_i^t|X_i)$, where

$$p(\mathbf{f}_i^v, \mathbf{f}_i^t|X_i) = p(\mathbf{f}_i^v|\mathbf{f}_i^t, X_i) \times p(\mathbf{f}_i^t|X_i^t), \tag{8}$$
$$= \mathcal{N}(\mathbf{f}_i^v|\mathbf{K}_i^{vt}[\mathbf{K}_i^t]^{-1}\mathbf{f}_i^t, \mathbf{K}_i^v - \mathbf{K}_i^{vt}[\mathbf{K}_i^t]^{-1}[\mathbf{K}_i^{vt}]^\top) \times \mathcal{N}(\mathbf{f}_i^t|\mathbf{0}, \mathbf{K}_i^t). \tag{9}$$

Here, $\mathbf{K}_i^t, \mathbf{K}_i^v$ are square kernel matrices of size $n^t \times n^t$ and $n^v \times n^v$ on the training inputs $X_i^t$ and validation inputs $X_i^v$, while $\mathbf{K}_i^{vt}$ is the $n^v \times n^t$ cross kernel matrix between these two sets of inputs. Note that the encoding is *non-parametric* since its size grows with the number of task data points.

To continue with the specification of the VIB objective, the second quantity we need to set is the decoder model $p_\theta(Y_i^v|\mathbf{f}_i^v, \mathbf{f}_i^t, X_i)$ which is chosen to be a standard GP likelihood. Specifically, for i.i.d. observations $Y_i^v$ given $\mathbf{f}_i^v$ becomes independent from $\mathbf{f}_i^t$ and $X_i$ and the likelihood factorizes across data points, i.e. $p(Y_i^v|\mathbf{f}_i^v) = \prod_{j=1}^{n^v} p(y_{i,j}^v|f_{i,j}^v)$, where each $p(y_{i,j}^v|f_{i,j}^v)$ is a standard likelihood model, such as Gaussian density $p(y_{i,j}^v|f_{i,j}^v) = \mathcal{N}(y_{i,j}^v|f_{i,j}^v, \sigma^2)$ suitable for standard regression problems or categorical/softmax likelihood for few-shot classification; see Appendix C.4. Finally, we specify the encoder distribution $q_w(Z_i|Y_i^t, X_i) \equiv q(\mathbf{f}_i^v, \mathbf{f}_i^t|Y_i^t, X_i)$ as follows,

$$q(\mathbf{f}_i^v, \mathbf{f}_i^t|Y_i^t, X_i) = p(\mathbf{f}_i^v|\mathbf{f}_i^t, X_i)q(\mathbf{f}_i^t|\mathcal{D}_i^t), \tag{10}$$

where $p(\mathbf{f}_i^v|\mathbf{f}_i^t|X_i)$ is the same conditional GP prior from equation 8, while $q(\mathbf{f}_i^t|\mathcal{D}_i^t)$ is a encoder of the training set that takes the form of a Gaussian distribution specified based on a very general amortization procedure, see Appendix C.1. Equation 10 shares a similar structure with a standard posterior Gaussian process where we first observe the training set, then we compute the (approximate) posterior distribution $q(\mathbf{f}_i^t|\mathcal{D}_i^t)$, and finally we extrapolate/predict the validation set function values at inputs $X_i^v$ based on the conditional GP prior $p(\mathbf{f}_i^v|\mathbf{f}_i^t, X_i)$. The above assumptions yield (see Appendix C.2) the following VIB objective for a single task,

$$\sum_{j=1}^{n^v} \mathbb{E}_{q(f_{i,j}^v)}[\log p(y_{i,j}^v|f_{i,j}^v)] - \beta \text{KL}\left[q(\mathbf{f}_i^t|\mathcal{D}_i^t)||p(\mathbf{f}_i^t|X_i^t)\right], \tag{11}$$

where $q(f_{i,j}^v) = \int p(f_{i,j}^v|\mathbf{f}_i^t, x_{i,j}^v, X_i^t)q(\mathbf{f}_i^t|\mathcal{D}_i^t)d\mathbf{f}_i^t$ is a univariate Gaussian over an individual validation function value $f_{i,j}^v$. Here, $q(f_{i,j}^v)$ depends on the training set and the single validation input $x_{i,j}^v$, so that from the training set and the corresponding function values $\mathbf{f}_i^t$ we extrapolate (through the univariate conditional GP $p(f_{i,j}^v|\mathbf{f}_i^t, x_{i,j}^v, X_i^t)$) to the input $x_{i,j}^v$ in order to predict its value $f_{i,j}^v$.

## 3.2 COMBINATION WITH GRADIENT-BASED META LEARNING

In this section, we combine the GP memory-based meta learning method with a gradient-based approach, such as MAML (Finn et al., 2017). Based on the VIB principle we need to specify an encoding that will allow us to combine a memory-based with a gradient-based approach. As discussed in Section 2.2, gradient-based meta learning is associated with a parametric encoding that corresponds to a fixed-size task model parameter $\psi_i$. In contrast, as seen in Section 3.1, a memory system is associated with a function-space or non-parametric encoding that consists of the

---

[3]The kernel variance parameter $\sigma_f^2$ (if learnable) is also considered to be part of the full set of parameters $\theta$.

function values $(\mathbf{f}_i^t, \mathbf{f}_i^v)$ of all task input points. Therefore, a way to combine these techniques is to concatenate the encodings, i.e. $Z_i \equiv (\psi_i, \mathbf{f}_i^t, \mathbf{f}_i^v)$. Here, $\psi_i$ are the task-specific parameters of the GP kernel function $k_{\psi_i}(x, x')$ (and possibly of the likelihood $p(y|f)$), which in our implementation are the parameters of the feature vector $\phi(x; \psi_i)$. Intuitively, a combination of the GP memory-based method with MAML will try to apply a short inner adaptation loop in order to adjust an initial feature vector $\phi(x; \theta)$ to obtain a final $\phi(x; \psi_i)$ that can better solve the task. For the overall encoding $(\psi_i, \mathbf{f}_i^t, \mathbf{f}_i^v)$, the general form of the encoder distribution takes the form

$$q(\mathbf{f}_i^t, \mathbf{f}_i^v, \psi_i | Y_i^t, X_i) = p(\mathbf{f}_i^v | \mathbf{f}_i^t, \psi_i, X_i) q(\mathbf{f}_i^t | \psi_i, \mathcal{D}_i^t) q(\psi_i | \mathcal{D}_i^t),$$

where $p(\mathbf{f}_i^v | \mathbf{f}_i^t, \psi_i, X_i)$ is the conditional GP prior and $q(\mathbf{f}_i^t | \psi_i, \mathcal{D}^t)$ is the amortized encoder (see Appendix C.1), where we have emphasized their dependence on $\psi_i$. The VIB objective becomes

$$\mathbb{E}_{q(\psi_i | \mathcal{D}_i^t)}\left[\sum_{j=1}^{n^v} \mathbb{E}_{q(f_{i,j}^v | \psi_i)}[\log p(y_{i,j}^v | f_{i,j}^v)] - \beta \mathrm{KL}[q(\mathbf{f}_i^t | \psi_i, \mathcal{D}_i^t) || p(\mathbf{f}_i^t | \psi_i, X_i^t)]\right] - \beta \mathrm{KL}[q(\psi_i | \mathcal{D}_i^t) || p(\psi_i)].$$

In practise, we can relax this objective and use different hyperparameters $\beta_f$ and $\beta_\psi$ in front of the two KL terms. This is convenient as we would prefer to set $\beta_\psi = 0$ and use a deterministic MAML w.r.t. $\psi_i$ rather than a stochastic MAML; see Section 2.2. This simplification avoids the need to specify a prior $p(\psi_i)$ over the task-specific neural network parameters and at the same time it reduces the encoder $q(\psi_i | \mathcal{D}_i^t)$ to a Dirac delta, which simplifies the objective as follows,

$$\sum_{j=1}^{n^v} \mathbb{E}_{q(f_{i,j}^v | \psi_i)}[\log p(y_{i,j}^v | f_{i,j}^v)] - \beta_f \mathrm{KL}\left[q(\mathbf{f}_i^t | \psi_i, \mathcal{D}_i^t) || p(\mathbf{f}_i^t | \psi_i, X_i^t)\right],$$

where $\psi_i = \theta + \Delta(\theta, \mathcal{D}_i^t)$. The inner loop adaptation term can be defined by an objective function on the support set $\mathcal{D}_i^t$. In our case a suitable objective is the VIB for the GP memory-based method obtained by setting the validation set equal to the training set, i.e. $D_i^v = D_i^t$ in equation 11, which gives $\sum_{j=1}^{n^t} \mathbb{E}_{q(f_{i,j}^t | \theta)}[\log p(y_{i,j}^t | f_{i,j}^t)] - \beta_f \mathrm{KL}\left[q(\mathbf{f}_i^t | \theta, \mathcal{D}_i^t) || p(\mathbf{f}_i^t | \theta, X_i^t)\right]$.

# 4 RELATED WORK

In this work, the VIB principle was used to formulate meta learning. VIB has been used before for different purposes, such as for regularization of single-task supervised learning (Alemi et al., 2017), sparse coding (Chalk et al., 2016), re-interpretation of $\beta$-VAEs and Dropout (Burgess et al., 2018; Achille & Soatto, 2016) and for compression of deep neural networks (Dai et al., 2018).

A meta learning method that connects with the information bottleneck was recently proposed by Hu et al. (2020). There the information bottleneck was used to analyze the generalization of a variational Bayesian inference objective suitable for transductive supervised few-shot learning. Note that the information bottleneck derived in Hu et al. (2020) (theorem 1) is not the same as the information bottleneck objective used here (i.e. the objective in equation 7 for the supervised learning case), since they differ in the second term. In addition, our framework expresses a general information bottleneck principle for meta learning applicable beyond transductive supervised learning.

Given the probabilistic nature of our framework, we can relate it to other probabilistic or Bayesian approaches and particularly with those that: (i) probabilistically re-interpret and extend gradient-based methods (Grant et al., 2018; Finn et al., 2018; Yoon et al., 2018; Nguyen et al., 2019; Gordon et al., 2019; Chen et al., 2019) and (ii) derive amortized conditional probability models (Garnelo et al., 2018; Gordon et al., 2019). The underlying learning principle in both (i)-(ii) is to construct and maximize a predictive distribution (or conditional marginal likelihood) of the validation points given the training points, which, e.g. in supervised few-shot learning is written as $p_\theta(Y^v | X^v, X^t, Y^t) = \int p(Y^v | X^v, \psi_i) p_\theta(\psi_i | X^t, Y^t) d\psi_i = \frac{p_\theta(Y^v, Y^t | X^v, X^t)}{p_\theta(Y^t | X^t)}$. Here, $p_\theta(\psi_i | X^t, Y^t)$ is a posterior distribution over the task parameters $\psi_i$, after observing the training points, and $\theta$ is a meta parameter which for simplicity we assume to be found by point estimation. However, this objective is very hard to rigorously approximate. Unlike the marginal likelihood on all task outputs $p_\theta(Y^v, Y^t | X^v, X^t)$ for which we can easily compute a lower bound, there is no tractable lower bound on the predictive conditional $p_\theta(Y^v | X^v, X^t, Y^t)$.[4] This inherent difficulty

---

[4]To obtain such a bound we either need to have access to the intractable posterior $p_\theta(\psi_i | X^t, Y^t)$, or to upper bound the marginal likelihood on the training points $p_\theta(Y^t | X^t)$, which is very hard.

with computing the predictive distribution has led to several approximations, i.e. methods of category (i) above, ranging from MAP, Laplace, variational inference procedures (without rigorous bounds on the predictive conditional) and Stein variational gradient descent (Grant et al., 2018; Finn et al., 2018; Yoon et al., 2018; Nguyen et al., 2019; Gordon et al., 2019; Chen et al., 2019). The conditional probability modelling approaches (Garnelo et al., 2018; Gordon et al., 2019) try to directly model $p_\theta(Y^v|X^v, X^t, Y^t)$ without considering this as an approximation to an initial joint Bayesian model. Our VIB framework differs significantly from the predictive distribution principle since VIB has an information theoretic motivation and it rigorously bounds an information bottleneck objective. VIB is also a fully tractable objective, avoiding the need to choose a particular approximate inference method and allowing us to rather focus on setting up the encoding procedure, as we did in the GP example in Section 3.

Regarding related works of GPs in meta learning, the ALPaCA method (Harrison et al., 2018) applied GPs and Bayesian linear regression to standard regression tasks with Gaussian likelihoods, while Tossou et al. (2019) used kernel-based methods (from a regularization rather than Bayesian perspective) again in standard regression. Closer to us, Patacchiola et al. (2019) and Snell & Zemel (2020) used GPs to perform few-shot classification together with deep neural kernels. However, our usage of GPs is different from these latter approaches, e.g. our general encoder amortization strategy can potentially deal with arbitrary likelihood functions and task output observations, while Patacchiola et al. (2019) assumes a Gaussian likelihood for the binary class labels and Snell & Zemel (2020) consider the Pólya-Gamma augmentation, which is very tailored to classification problems.

## 5 EXPERIMENTS

To evaluate the proposed algorithms, we consider a standard set of meta-learning benchmarks which include sinusoid regression and few-shot classification. As a baseline for comparison we consider MAML (Finn et al., 2017) and use exactly the same neural architecture for all methods and benchmarks as in Finn et al. (2017). The new methods we implemented are the following: (i) Stochastic MAML (S. MAML), as defined in Section 2.2, where the difference with MAML is the injected task-specific noise added to the outer loop update rule, together with the regularization term that comes from the VIB objective; see Section 2.2. The scalar noise parameter $s$ is learned alongside the others. (ii) The memory-based GP method (referred as GP) trained by the VIB objective, as defined in Section 3, where the kernel feature vector $\phi(x; \theta)$ is obtained from the last hidden layer of the same neural architecture as used in MAML. We use cosine and linear kernels (see Appendix D). For few-shot classification we report results separately for both kernels, while for sinusoid regression we only consider the linear kernel (the cosine kernel has similar performance). (iii) A combination of a memory-based and gradient-based approach referred to as GP+MAML where a MAML meta update rule is applied to the feature vector parameters $\theta$; see Section 3.2.

**Sinusoid regression.** We first evaluate the method on a sinusoid regression as described by Finn et al. (2017) following their protocol. We train GP using VIB and evaluate its K-shot mean squared error performance. From Table 1, we observe that GP significantly outperforms MAML especially as $K$ grows. Similarly, Figure 3 in Appendix illustrates this few-shot predictive ability of the GP, where posterior GP uncertainty reduces as $K$ grows. Finally, from Figure 2 (left), we observe that GP drastically improves when more data is available and is more data efficient than MAML achieving near-optimal performance for $K = 5$. See Appendix D for more results and details.

**Table 1:** Few-shot sinusoid regression results. We report the results (mean values and 95% confidence intervals after 10 repeats) of all methods (GP, GP+MAML, MAML and S. MAML) for $K = 5, 10, 20$. GP-based methods use linear kernel.

| Model | K=5 | K=10 | K=20 |
|---|---|---|---|
| MAML | 0.280±0.013 | 0.096±0.005 | 0.043±0.003 |
| S. MAML | 0.317±0.34 | 0.116±0.012 | 0.054±0.004 |
| GP | 0.02±0.014 | 0.002±0.001 | 0.001±0.001 |
| GP+MAML | 0.058±0.054 | 0.002±0.001 | 0.002±0.002 |

**Few-shot classification.** The second domain is a standard few-shot meta-learning benchmark based on three datasets: Omniglot (Lake et al., 2011), mini-Imagenet (Ravi & Larochelle, 2017) and Augmented Omniglot (Flennerhag et al., 2018). For both Omniglot and mini-Imagenet we follow the experimental protocol proposed by Finn et al. (2017). For Augmented Omniglot, following Flennerhag et al. (2018); Chen et al. (2019), during meta-testing both MAML and S. MAML perform

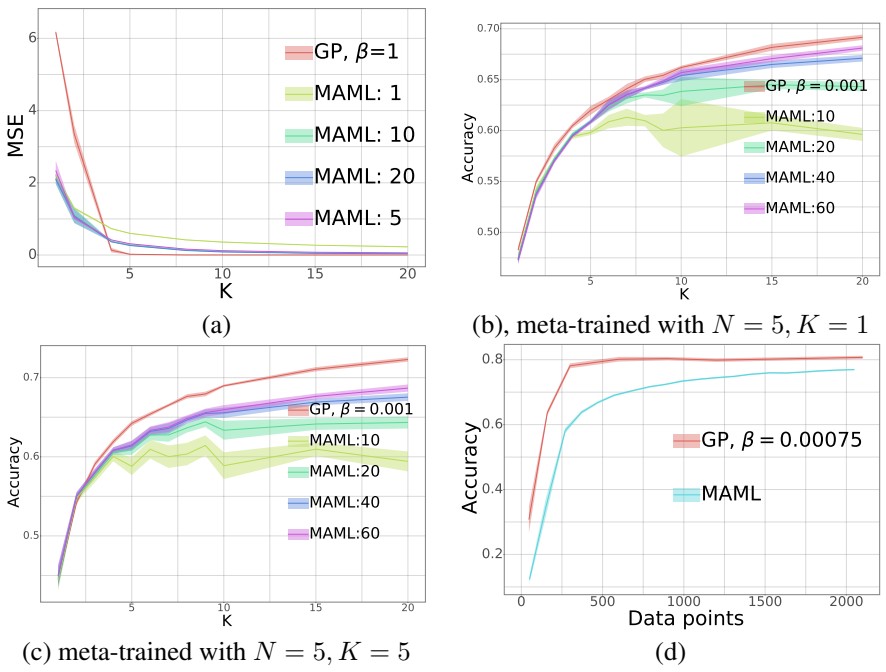

**Figure 2:** (**a**) Sinusoid regression with GP and MAML in meta-testing as the number of shots $K$ (x-axis) increases. On y-axis, we report the MSE. For MAML, we report the performance with different number of inner loop steps, i.e. just SGD steps since we do meta-testing, specified in the legend. (**b**)-(**c**) Meta-testing classification accuracy (y-axis) on mini-Imagenet, where each system has been meta-trained with either $N = 5, K = 1$ or $N = 5, K = 5$, as the number $K$ of observed examples per class (while always $N = 5$) grows from 1 to 20, e.g. for $K = 20$ each system sees $N \times K = 100$ support examples. For MAML we show the performance for different inner loop sizes, which in meta-testing is just SGD updates, where each SGD step uses a random mini-batch of size 10 data points from the support set of size $N \times K$. (**d**) Similarly to (**b**)-(**c**) for Augmented Omniglot, where instead of growing $K$ we increase the amount of data augmentation in a pre-specified/fixed $N = 20, K = 15$ initial support set. This means in each SGD update of MAML or predictive density GP update we sample a mini-batch from the fixed $N \times K$ support set, we apply random transformations in this mini-batch and then we use it to perform the actual update. The mini-batch size was 20 and based on this data augmentation process we grow the amount of data (x-axis) processed by each method from 20 up to 2000. Finally to create all plots we average performance under 10 repeats, where in each repeat the systems are meta-trained from scratch and then are evaluated in a large number of meta-testing tasks.

| Model | **Omniglot** 5-way | | **mini-Imagenet** 5-way | | **Augmented Omniglot** 20-way |
|---|---|---|---|---|---|
| | $K = 1$ shot | $K = 5$ shot | $K = 1$ shot | $K = 5$ shot | $K = 15$ shot |
| Stochastic MAML | 98.944±0.031% | 99.745±0.017% | 48.139±0.311% | 64.179±0.781% | 77.3±1.123% |
| GP (linear) | 99.036±0.034% | 99.776±0.019% | 48.097±0.213% | 64.474±0.235% | 80.73±0.776% |
| GP (cos) | **99.059**±0.030% | 99.774±0.016% | 47.934±0.293% | 63.660±0.262% | 80.77±0.804% |
| GP+MAML (linear) | 99.027±0.025% | 99.771±0.019% | 48.509±0.346% | **64.760**±0.226% | 81.33±0.488% |
| GP+MAML (cos) | 99.051±0.028% | 99.783±0.019% | 48.075±0.445% | 64.506±0.178% | *__81.79__±0.671% |
| MAML (Finn et al., 2017) | 98.7±0.4% | **99.9**±0.1% | **48.7**±1.84% | 63.11±0.92% | 76.67±0.663% (our) |

**Table 2:** Classification test accuracy on Omniglot, mini-Imagenet and Augmented Omniglot. For all methods, mean performances with 95% confidence intervals are reported after repeating the experiments 10 times and each experiments performs meta-testing in 1000 tasks. Best performance is with bold, while * indicates statistically significant better performance than MAML in Augmented Omniglot. For Augmented Omniglot we run MAML (see Appendix for the hyperparameters) since this dataset was not included in Finn et al. (2017).

100 steps of adaptation (resulting in 2000 data points seen by the model where each step processes a minibatch of size 20 points), while they are meta-trained by applying 20 adaptation steps (i.e. 400 training points seen per task). Both GP methods are meta-trained by memorizing the full $N \times K = 20 \times 15 = 300$ support points without further data augmentation, while during meta-testing we allow the GP methods to see up to 2000 points. See Appendix D for more details.

| | **Omniglot** 5-way | | **mini-Imagenet** 5-way | | **Augmented Omniglot** 20-way |
|---|---|---|---|---|---|
| **Model** | $K = 1$ shot | $K = 5$ shot | $K = 1$ shot | $K = 5$ shot | $K = 15$ shot |
| Stochastic MAML | **0.031**±0.001% | **0.008**±0.001 | 1.27±0.008% | 0.925±0.013% | 0.673±0.025% |
| GP (linear) | 0.036±0.002% | 0.012±0.001% | 1.267±0.008% | 0.904±0.005% | 0.676±0.034% |
| GP (cos) | 0.045±0.001% | 0.019±0.001% | 1.262±0.006% | 0.921±0.006% | 0.662±0.027% |
| GP+MAML (linear) | 0.036±0.003% | 0.010±0.001% | *1.246±0.007% | *0.900±0.009% | 0.671±0.024% |
| GP+MAML (cos) | 0.045±0.001% | 0.019±0.001% | 1.274±0.009% | 0.902±0.005% | *0.616±0.027% |
| MAML (our) | 0.032±0.001% | **0.008**±0.001% | 1.279±0.006% | 0.926±0.011% | 0.694±0.02% |

**Table 3:** Classification negative log likelihood (NLL) test performance on Omniglot, mini-Imagenet and Augmented Omniglot. To obtain these scores for MAML we re-run MAML since only classification accuracy is reported in Finn et al. (2017). Again * indicates statistically significant better performance than MAML.

Classification accuracy performance for all methods are given in Table 2, while the corresponding negative log likelihood (NLL) scores are given in Table 3. We observe GP-based architectures outperform MAML and S. MAML in more complex scenarios such as Augmented Omniglot. From the NLL scores that depend on how well the predicted class probabilities are calibrated, we can observe that the GP methods perform significantly better in all cases where uncertainty matters i.e. on mini-Imagenet and Augmented Omniglot. On top of the described standard implementations of GP and GP+MAML, we implement ones with additional previously used architectures and tricks to improve results (see Appendx D.4). We observe that GP+MAML generally performs better than GP. Further, we notice that S. MAML has similar performance to MAML. One reason why S. MAML does not always outperform MAML could be related to hyperparameters and to the higher variance of the gradients caused by the reparametrization trick used to maximize the VIB objective in equation 7. This could imply that training a stochastic architecture could require longer training time, an issue deserving further investigation.

We also found that the qualitative behaviour of GP and GP+MAML are quite different as shown in Figure 4 in the Appendix D.2. In Appendix D.3 we provide ablative analysis for the impact of $\beta$ on the performance of our architecture where the main result is that a large range of $\beta$ values gives similar performance in practice.

**Data efficiency in meta-testing of gradient-based vs GP-based meta learning.** Having performed an expensive meta-training phase, the ultimate goal of a meta-learning system is to be deployed in practice and solve new tasks. In a real meta-testing scenario it is more likely the system to operate in a regime, where the training data for a given task is coming sequentially in mini-batches, and the system should continuously update itself. With this in mind, it is interesting to study the effect of $K$ shots (or more generally the effect of the amount of processed data) in meta-testing performance. In Figure 2, we carry out an ablation study by comparing MAML and GP in sinusoid regression, mini-Imagenet and Augmented Omniglot by varying either $K$, for sinusoid regression and mini-Imagenet, or the amount of data augmentation, for Augmented Omniglot. We found that GP can be much more data efficient than MAML. This is because the GP predictive updates are based on Bayesian updating of sufficient statistics (for the linear kernels used here) which are similarly to Bayesian linear regression, as described in detail in Appendix C.3. Such updates are exchangeable (i.e. data order does not matter on the final prediction) and do not depend on learning rates. In contrast, a gradient-based method such as MAML can be less data efficient since stochastic gradient updates depend on the learning rate, mini-batch size, data order and the size of the inner loop.

## 6 Conclusion

We introduced an information theoretic framework for meta learning by using a variational approximation (Alemi et al., 2017; Chalk et al., 2016; Achille & Soatto, 2016) to the information bottleneck (Tishby et al., 1999). We derived a novel memory-based meta learning method with GPs, a stochastic MAML and a combination of memory and gradient-based meta learning.

While we have demonstrated our method in few-shot regression and classification, we believe that the scope of the information bottleneck for meta learning is much broader. For instance, an interesting topic for future research is to consider applications in reinforcement learning.

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

Here, we provide additional details regarding our method. Appendix A describes fully the derivation of the variational information bottleneck (VIB) objective for meta learning. Appendix B explains how the transductive and non-transductive settings, often used in few-shot image classification, can be explained as a particular cases of the VIB framework under suitably defined encodings. Appendix C provides full details regarding the GP meta learning method proposed in the main paper. Finally, Appendix D discusses experimental settings, and it presents additional experimental results and ablation studies.

## A    FURTHER DETAILS ABOUT VIB IN META LEARNING

### A.1    BOUNDS ON THE MUTUAL INFORMATION

Here, we review the standard variational bounds on the mutual information from Barber & Agakov (2003). Recall the definition of the mutual information,

$$I(x, y) = \int q(x, y) \log \frac{q(x, y)}{q(x)q(y)} dxdy = \int q(x, y) \log \frac{q(x|y)}{q(x)} dxdy.$$

By introducing $p(x|y)$ that approximates $q(x|y)$ we get

$$I(x, y) = \int q(x, y) \log \frac{p(x|y)q(x|y)}{p(x|y)q(x)} dxdy = \int q(x, y) \log \frac{p(x|y)}{q(x)} dxdy + \int q(y) \mathrm{KL}[q(x|y)||p(x|y)]dy,$$

which shows that

$$I(x, y) \geq \int q(x, y) \log \frac{p(x|y)}{q(x)} dxdy, \tag{12}$$

since $\int q(y) \mathrm{KL}[q(x|y)||p(x|y)]dy$ is non negative. An upper bound is obtained similarly. Suppose $p(x)$ approximates $q(x)$ then

$$I(x, y) = \int q(x, y) \log \frac{p(x)q(x|y)}{p(x)q(x)} dxdy = \int q(x, y) \log \frac{q(x|y)}{p(x)} dxdy - \mathrm{KL}[q(x)||p(x)]dy,$$

which shows that

$$I(x, y) \leq \int q(x, y) \log \frac{q(x|y)}{p(x)} dxdy. \tag{13}$$

### A.2    THE GENERAL VIB META LEARNING CASE

Consider the general case, where we work with the unconditional mutual information and we wish to approximate the IB: $I(Z, \mathcal{D}^v) - \beta I(Z, \mathcal{D}^t)$. Recall that the joint distribution is written as

$$q_w(\mathcal{D}^v, \mathcal{D}^t, Z) = q_w(Z|\mathcal{D}^t)p(\mathcal{D}^v, \mathcal{D}^t), \tag{14}$$

from which we can express any marginal or conditional. In particular observe that

$$q_w(Z, D^v) = \int q_w(Z|\mathcal{D}^t)p(\mathcal{D}^v, \mathcal{D}^t)d\mathcal{D}^t.$$

If we have a function $f(Z, D^v)$ and we wish to approximate the expectation,

$$\int q_w(Z, \mathcal{D}^v)f(Z, \mathcal{D}^v)dZd\mathcal{D}^v = \int q_w(Z|\mathcal{D}^t)p(\mathcal{D}^v, \mathcal{D}^t)f(Z, D^v)dZd\mathcal{D}^vd\mathcal{D}^t, \tag{15}$$

then given that we sample a task pair $(\mathcal{D}_i^v, \mathcal{D}_i^t) \sim p(\mathcal{D}^v, \mathcal{D}^t)$ we can obtain the following unbiased estimate of this expectation,

$$\int q_w(Z|\mathcal{D}_i^t)f(Z, \mathcal{D}_i^v)dZ. \tag{16}$$

We are going to make use of equation 15 and equation 16 in the derivation below.

To compute the variational approximation to IB, we need to lower bound $I(Z, \mathcal{D}^v)$ as

$$
\begin{aligned}
I(Z, \mathcal{D}^v) &= \int q_w(Z, \mathcal{D}^v) \log \frac{q_w(Z, \mathcal{D}^v)}{q_w(Z)p(\mathcal{D}^v)} = \int q_w(Z, \mathcal{D}^v) \log \frac{q_w(\mathcal{D}^v|Z)}{p(\mathcal{D}^v)} dZ d\mathcal{D}^v \\
&\geq \int q_w(Z, \mathcal{D}^v) \log \frac{p_\theta(\mathcal{D}^v|Z)}{p(\mathcal{D}^v)} dZ d\mathcal{D}^v, \quad \text{by using equation 12} \\
&= \int q_w(Z, \mathcal{D}^v) \log p_\theta(\mathcal{D}^v|Z) dZ d\mathcal{D}^v + \mathcal{H}(\mathcal{D}^v),
\end{aligned}
$$

where the entropy $\mathcal{H}(\mathcal{D}^v)$ is just a constant.

Subsequently, we upper bound $I(Z, \mathcal{D}^t)$ as follows,

$$
\begin{aligned}
I(Z, \mathcal{D}^t) &= \int q_w(Z, \mathcal{D}^t) \log \frac{q_w(Z, \mathcal{D}^t)}{q_w(Z)p(\mathcal{D}^t)} dZ d\mathcal{D}^t = \int q_w(Z|\mathcal{D}^t)p(\mathcal{D}^t) \log \frac{q_w(Z|\mathcal{D}^t)}{q_w(Z)} dZ d\mathcal{D}^t, \\
&\leq \int q_w(Z|\mathcal{D}^t)p(\mathcal{D}^t) \log \frac{q_w(Z|\mathcal{D}^t)}{p_\theta(Z)} dZ d\mathcal{D}^t \quad \text{by using equation 13}
\end{aligned}
$$

Then we obtain the overall loss, $\mathcal{F}(\theta, w) \leq \mathcal{L}_{IB}(w)$:

$$
\mathcal{F}(\theta, w) = \int q_w(Z, \mathcal{D}^v) \log p_\theta(\mathcal{D}^v|Z) dZ d\mathcal{D}^v - \beta \int q_w(Z|\mathcal{D}^t)p(\mathcal{D}^t) \log \frac{q_w(Z|\mathcal{D}^t)}{p_\theta(Z)} dZ d\mathcal{D}^t,
$$

where we dropped the constant entropic term $\mathcal{H}(\mathcal{D}^v)$. Therefore, given a set of task pairs $\{\mathcal{D}_i^t, \mathcal{D}_i^v\}_{i=1}^b$, where each $(\mathcal{D}_i^t, \mathcal{D}_i^v) \sim \widetilde{p}(\mathcal{D}^v, \mathcal{D}^t)$, the objective function for learning $(\theta, w)$ becomes the empirical average, $\frac{1}{b} \sum_{i=1}^b \mathcal{F}_i(\theta, w)$, where

$$
\mathcal{F}_i(w, \theta) = \int q_w(Z_i|\mathcal{D}_i^t) \log p_\theta(\mathcal{D}_i^v|Z_i) dZ_i - \beta \int q_w(Z_i|\mathcal{D}_i^t) \log \frac{q_w(Z_i|\mathcal{D}_i^t)}{p_\theta(Z_i)} dZ_i, \tag{17}
$$

where for the first term we made use of equation 15 and equation 16 with $f(\mathcal{D}^v, Z) = \log p_\theta(\mathcal{D}^v|Z)$.

## A.3 THE SUPERVISED META LEARNING VIB CASE

For the supervised meta learning case the joint density can be written as

$$
\begin{aligned}
q_w(\mathcal{D}^v, \mathcal{D}^t, Z) &= q_w(Z|Y^t, X^t, X^v)p(Y^t, Y^v|X^t, X^v)p(X^v, X^t), \\
&= q_w(Z|Y^t, X)p(Y^t, Y^v|X)p(X), \tag{18}
\end{aligned}
$$

where $X = (X^t, X^v)$ and the encoding distribution $q_w(Z|Y^t, X)$ could depend on all inputs $X$ but only on the training outputs $Y^t$. The derivation of the VIB objective is similar with the general case with the difference that now we approximate the conditional information bottleneck $I(Z, Y^v|X) - \beta I(Z, Y^t|X)$ where we condition on the inputs $X$. In other words, both $I(Z, Y^v|X)$ and $I(Z, Y^t|X)$ are conditional mutual informations, i.e. they have the form

$$
I(z, y|x) = \int q(x) \left[ \int q(z, y|x) \log \frac{q(z, y|x)}{q(z|x)q(y|x)} dz dy \right] dx = \int q(z, y, x) \log \frac{q(z, y|x)}{q(z|x)q(y|x)} dz dy dx.
$$

We can lower bound $I(Z, Y^v|X)$, as follows,

$$
\begin{aligned}
\int p(X) &\left[ \int q_w(Z, Y^v|X) \log \frac{q_w(Z, Y^v|X)}{q_w(Z|X)p(Y^v|X)} dZ dY^v \right] dX \\
&= \int p(X) \int q_w(Z, Y^v|X) \log \frac{q_w(Y^v|Z, X)}{p(Y^v|X)} dZ dY^v dX \quad \text{where } q_w(Z|X) \text{ cancels} \\
&\geq \int p(X) \int q_w(Z, Y^v|X) \log \frac{p_\theta(Y^v|Z, X)}{p(Y^v|X)} dZ Y^v dX \quad \text{by using equation 12} \\
&= \int q_w(Z, Y^v, X) \log \frac{p_\theta(Y^v|Z, X)}{p(Y^v|X)} dZ dY^v dX \\
&= \int q_w(Z, Y^v, X) \log p_\theta(Y^v|Z, X) dZ dY^v dX - \int p(Y^v, X) \log p(Y^v|X) dY^v dX \tag{19}
\end{aligned}
$$

Note that $-\int p(Y^v, X) \log p(Y^v|X) dY^v dX$ is just a constant that does not depend on tunable parameters. Also

$$q_w(Z, Y^v, X) = \int q_w(Z|Y^t, X) p(Y^t, Y^v|X) p(X) dY^t, \tag{20}$$

so that if we have a task sample $(Y_i^t, Y_i^v, X_i) \sim p(Y^t, Y^v|X) p(X)$ an unbiased estimate of the expectation $\int q_w(Z, Y^v, X) \log p_\theta(Y^v|Z, X) dZ dY^v dX$ is given by

$$\int q_w(Z|Y_i^t, X_i) \log p_\theta(Y_i^v|Z, X_i) dZ. \tag{21}$$

We upper bound $I(Z, Y^t|X)$ as follows,

$$\int p(X) \left[ \int q_w(Z, Y^t|X) \log \frac{q_w(Z, Y^t|X)}{q_w(Z|X) p(Y^t|X)} dZ dY^t \right] dX$$

$$= \int p(X) \left[ \int q_w(Z, Y^t|X) \log \frac{q_w(Z|Y^t, X)}{q_w(Z|X)} dZ dY^t \right] dX, \quad \text{where } p(Y^t|X) \text{ cancels}$$

$$\leq \int p(X) \int q_w(Z, Y^t|X) \log \frac{q_w(Z|Y^t, X)}{p_\theta(Z|X)} dZ dY^t dX, \quad \text{by using equation 13}$$

$$= \int q_w(Z|Y^t, X) p(Y^t, X) \log \frac{q_w(Z|Y^t, X)}{p_\theta(Z|X)} dZ dY^t dX,$$

Then we obtain the overall objective,

$$\mathcal{F}(\theta, w) = \int q_w(Z, Y^v, X) \log p_\theta(Y^v|Z, X) dZ dY^v dX$$

$$- \beta \int q_w(Z|Y^t, X) p(Y^t, X) \log \frac{q_w(Z|Y^t, X)}{p_\theta(Z|X)} dZ dY^t dX,$$

where we dropped the constant term. Therefore, given a set of task pairs the objective becomes the empirical average, $\frac{1}{b} \sum_{i=1}^{b} \mathcal{F}_i(\theta, w)$, where

$$\mathcal{F}_i(\theta, w) = \int q_w(Z|Y_i^t, X_i) \log p_\theta(Y_i^v|Z, X_i) dZ - \beta \int q_w(Z|Y_i^t, X_i) \log \frac{q_w(Z|Y_i^t, X_i)}{p_\theta(Z|X_i)} dZ, \tag{22}$$

where we made use of equation 21.

### A.4 CONNECTION WITH VARIATIONAL INFERENCE

As mentioned in the main paper, the VIB for meta learning (where we consider for simplicity the general case from A.2) is similar to applying approximate variational inference to a certain joint model over the validation set,

$$p_\theta(\mathcal{D}^v|Z) p_\theta(Z),$$

where $p_\theta(\mathcal{D}^v|Z)$ is the decoder model, $p_\theta(Z)$ a prior model over the latent variables and where the corresponding marginal likelihood is

$$p(\mathcal{D}^v) = \int p_\theta(\mathcal{D}^v|Z) p_\theta(Z) dZ.$$

We can lower bound the log marginal likelihood with a variational distribution $q_w(Z|\mathcal{D}^t)$ that depends on the training set $\mathcal{D}^t$,

$$\mathcal{F}_{\beta=1}(w, \theta) = \int q_w(Z|\mathcal{D}^t) \log p_\theta(\mathcal{D}^v|Z) dZ - \int q_w(Z|\mathcal{D}^t) \log \frac{q_w(Z|\mathcal{D}^t)}{p_\theta(Z)} dZ, \tag{23}$$

which corresponds to the VIB objective with $\beta = 1$.

## B    Transductive and non-transductive meta learning

Here, we discuss how the transductive and non-transductive settings that appear in few-shot image classification (Bronskill et al., 2020; Finn et al., 2017; Nichol et al., 2018), due to the use of batch-normalization, can be interpreted under our VIB framework by defining suitable encodings. We shall use MAML as an example, but the discussion is more generally relevant.

The transductive case occurs when the concatenated support and validation/test inputs $X = (X^t, X^v)$ of a single task (we ignore the task index $i$ to keep the notation uncluttered) are used to compute batch-norm statistics (possibly at different stages) shared by all validation/test points, when predicting those points. For MAML this implies a deterministic parametric encoding, i.e. common to all individual validation inputs $x_j^v \in X^v$, obtained by a sequence of two steps: (i) Obtain first the task-specific parameter $\psi$ in the usual way by the support loss, i.e. $\psi = \theta + \Delta(\theta, \mathcal{D}^t)$. If batch-normalization is used here, then the statistics are computed only by $X^t$. (ii) Compute the validation loss by applying batch-normalization on $X^v$ or the union $X = X^t \cup X^v$ (the union seems to be a better choice, but not used often in practice for computational reasons, e.g. Finn et al. (2017); Nichol et al. (2018) prefer to use only $X^v$). In both cases, the underlying encoder is parametric over the final effective task parameter $\widetilde{\psi} = BN(\psi, X)$, where $BN$ denotes the final batch-norm operation that outputs a parameter vector, that predicts all validation points and it is a deterministic delta measure.

In contrast, the non-transductive setting occurs when each individual validation input $x_j^v$ is concatenated with the support inputs $X^t$ to form the sets $x_j^v \cup X^t$, $j = 1, \ldots, n^v$. Then, each set $x_j^v \cup X^t$ is used to compute point-specific batch-norm statistics when predicting the corresponding validation output $y_j^v$. Under the VIB framework this corresponds to a non-parametric encoding, which grows with the size of the validation set. The first deterministic step of this encoder is the same (i) above from the transductive case but the second step differs in the sense that now we get a validation point-specific task parameter $\widetilde{\psi}_j = BN(\psi, x_j^v \cup X^t)$ by computing the statistics using the set $x_j^v \cup X^t$. For MAML, this encoding becomes, $Z \equiv \{\widetilde{\psi}_j\}_{j=1}^{n^v}$, and the encoder distribution is a product of delta measures. i.e. $p(\{\widetilde{\psi}_j\}_{j=1}^{n^v}|Y^t, X) \equiv \prod_{j=1}^{n^v} \delta_{\widetilde{\psi}_j, BN(\theta+\Delta(\theta, \mathcal{D}^t), x_j^v \cup X^t)}$.

Finally, note that under the VIB perspective it does not make much sense to meta-train transductively and meta-test non-transductively and via versa, since this changes the encoding. That is, whatever we do in meta-training we should do the same in meta-testing.

## C    Further details about the Gaussian process method

For simplicity next we ignore the task index $i$ to keep the notation uncluttered, and write for example $\mathbf{f}_i^t$ as $\mathbf{f}^t$ and etc.

### C.1    Amortization of the GP Encoder $q(\mathbf{f}^t|\mathcal{D}^t)$

A suitable choice of $q(\mathbf{f}^t|\mathcal{D}^t)$ is to set it equal to the exact posterior distribution over $\mathbf{f}^t$ given the training set, i.e. $p(\mathbf{f}^t|\mathcal{D}^t) \propto \prod_{j=1}^{n^t} p(y_j^t|f_j^t)\mathcal{N}(\mathbf{f}^t|0, \mathbf{K}^t)$. Interestingly, such a setting does not require to introduce any extra variational parameters $w$ and it will depend only on the model parameters $\theta$ that appear in the kernel function and possibly also in the likelihood. For standard regression problems where the likelihood is Gaussian, i.e. $p(y_j^t|f_j^t) = \mathcal{N}(y_j^t|f_j^t, \sigma^2)$, the exact posterior has an analytic form given by

$$p(\mathbf{f}^t|\mathcal{D}^t) = \mathcal{N}(\mathbf{f}^t|\mathbf{K}^t(\mathbf{K}^t + \sigma^2 I)^{-1}Y^t, \mathbf{K}^t - \mathbf{K}^t(\mathbf{K}^t + \sigma^2 I)^{-1}\mathbf{K}^t) \tag{24}$$

and thus we can set $q(\mathbf{f}^t|\mathcal{D}^t) = p(\mathbf{f}^t|\mathcal{D}^t)$. For all other cases where the likelihood is not Gaussian we need to construct an amortized encoding distribution by approximating each non-Gaussian likelihood term $p(y_j^t|f_j^t)$, with a Gaussian term similarly to how we often parametrize variational Bayes or Expectation-Propagation Gaussian approximation to a GP model (Hensman et al., 2014; Opper & Archambeau, 2009; Rasmussen & Williams, 2006) i.e.

$$p(y_j^t|f_j^t) \approx \mathcal{N}(m_j^t|f_j^t, s_j^t),$$

where $m_j^t \equiv m_w(y_j^t, x_j^t) \in \mathbb{R}$ and $s_j^t \equiv s_w(x_j^t) \in \mathbb{R}_+$ are neural network amortized functions that depend on tunable parameters $w$ and receive as input an individual data point $(y_j^t, x_j^t)$ associated with the latent variable $f_j^t$. We made the simplification that the output point might only influence the real-valued mean $m_w(y_j^t, x_j^t)$, while the variance $s_w(x_j^t)$ can depend only on the input. Based on the above the amortized encoder is a fully dependent multivariate Gaussian distribution having the form

$$q(\mathbf{f}^t|\mathcal{D}^t) = \mathcal{N}(\mathbf{f}^t|\mathbf{K}^t(\mathbf{K}^t + \mathbf{S}^t)^{-1}\mathbf{m}^t, \mathbf{K}^t - \mathbf{K}^t(\mathbf{K}^t + \mathbf{S}^t)^{-1}\mathbf{K}^t), \quad (25)$$

where $\mathbf{S}^t$ is a diagonal covariance matrix with the vector $(s_1^t, \ldots, s_{n^t}^t)$ in the diagonal and $\mathbf{m}^t$ is the vector of values $(m_1^t, \ldots, m_{n^t}^t)$. This allows to re-write the VIB objective in equation 11 in the following computationally more convenient form (see Appendix C.2 next):

$$\sum_{j=1}^{n^v} \mathbb{E}_{q(f_j^v)}[\log p(y_j^v|f_j^v)] - \beta \sum_{j=1}^{n^t} \mathbb{E}_{q(f_j^t)}[\log \mathcal{N}(m_j^t|f_j^t, s_j^t)] + \beta \log \mathcal{N}(\mathbf{m}^t|0, \mathbf{K}^t + \mathbf{S}^t), \quad (26)$$

where each marginal Gaussian distribution $q(f_j)$ when $x_j$ is either from the validation or the training set (or any other further test set) is computed by the same expression, $q(f_j) = \mathcal{N}(f_j|\mathbf{k}_j^t(\mathbf{K}^t + \mathbf{S}^t)^{-1}\mathbf{m}^t, k_j - \mathbf{k}_j^t(\mathbf{K}^t + \mathbf{S}^t)^{-1}\mathbf{k}_j^{t\top})$, where $\mathbf{k}_j^t \equiv k(x_j, X^t)$ is the $n^t$ dimensional row vector of kernel values between $x_j$ and the training inputs $X^t$ and $k_j \equiv k(x_j, x_j)$.

**Classification.** Here, we discuss how the above general amortization procedure can be particularized to classification problems, which is the standard application in few-shot learning. For notational simplicity we focus on binary classification, while multi-class classification is fully covered later in Appendix C.4.

Suppose a meta learning problem where each task is a binary classification problem where the binary class labels are encoded in $\{-1, 1\}$. To apply the method we simply need to specify the form of the amortized mean function $m_w(y_j^t, x_j^t)$ (recall that $s_w(x_j^t)$ is independent from the output $y_j^t$), which is chosen to be

$$m(y_j^t, x_j^t) = y_j^t \times \widetilde{m}_w(x_j^t),$$

where $\widetilde{m}_w(x_j^t)$ is a real-valued function given by the neural network. Notice that the dependence on the output label $y_j^t \in \{-1, 1\}$ simply changes the sign of $\widetilde{m}_w(x_j^t)$. This latter function acts as a discriminative function that should tend towards positive values for data from the positive class and negative values for data from the negative class, while the product $y_j^t \times \widetilde{m}_w(x_j^t)$ should tend towards positive values. This amortization of the mean function is invariant to class re-labeling, i.e. if we swap the roles of the two labels $\{-1, 1\}$ the amortization remains valid and it does not require any change. The multi-class classification case can be dealt with similarly, by introducing as many latent functions as classes, as discussed fully in Appendix C.4.

## C.2 DERIVATION OF THE VIB BOUND

The VIB objective for a single task from Eq. equation 22 in the main paper is computed as follows

$$\sum_{j=1}^{n^v} \mathbb{E}_{q(f_j^v)}[\log p(y_j^v|f_j^v)] - \beta \int p(\mathbf{f}^v|\mathbf{f}^t, X^v, X^t)q(\mathbf{f}^t|\mathcal{D}^t) \log \frac{p(\mathbf{f}^v|\mathbf{f}^t, X^v, X^t)q(\mathbf{f}^t|\mathcal{D}^t)}{p(\mathbf{f}^v|\mathbf{f}^t, X^v, X^t)p(\mathbf{f}^t|X^t)} d\mathbf{f}^t d\mathbf{f}^v$$

$$\sum_{j=1}^{n^v} \mathbb{E}_{q(f_j^v)}[\log p(y_j^v|f_j^v)] - \beta \int q(\mathbf{f}^t|\mathcal{D}^t) \log \frac{q(\mathbf{f}^t|\mathcal{D}^t)}{p(\mathbf{f}^t|X^t)} d\mathbf{f}^t$$

$$\sum_{j=1}^{n^v} \mathbb{E}_{q(f_j^v)}[\log p(y_j^v|f_j^v)] - \beta \mathrm{KL}\left[q(\mathbf{f}^t|\mathcal{D}^t)||p(\mathbf{f}^t|X^t)\right], \quad (27)$$

where $q(f_j^v) = \int p(f_j^v|\mathbf{f}^t, x_j^v, X^t)q(\mathbf{f}^t|\mathcal{D}^t)d\mathbf{f}^t$ is a marginal Gaussian over an individual validation function value $f_j^v$, as also explained in the main paper. Specifically, $q(f_j^v)$ depends on the training set $(Y^t, X^t)$ and the single validation input $x_j^v$, so intuitively from the training set and the corresponding function values $\mathbf{f}^t$ we extrapolate (through the conditional GP $p(f_j^v|\mathbf{f}^t, x_j^v, X^t)$) to the input $x_j^v$ in order to predict its function value $f_j^v$.

Given the specific amortization of $q(\mathbf{f}^t|\mathcal{D}^t)$:

$$q(\mathbf{f}^t|\mathcal{D}^t) = \frac{\left(\prod_{j=1}^{n^t} \mathcal{N}(m_j^t|s_j^t)\right)\mathcal{N}(\mathbf{f}^t|0,\mathbf{K}^t)}{\mathcal{N}(\mathbf{m}^t|0,\mathbf{K}^t+\mathbf{S}^t)} = \mathcal{N}(\mathbf{f}^t|\mathbf{K}^t(\mathbf{K}^t+\mathbf{S}^t)^{-1}\mathbf{m}^t,\mathbf{K}^t-\mathbf{K}^t(\mathbf{K}^t+\mathbf{S}^t)^{-1}\mathbf{K}^t),$$

(28)

the VIB objective, by using the middle part of equation 28, can be written in the following form,

$$\sum_{j=1}^{n^v}\mathbb{E}_{q(f_j^v)}[\log p(y_j^v|f_j^v)] - \beta\sum_{j=1}^{n^t}\mathbb{E}_{q(f_j^t)}[\log\mathcal{N}(m_j^t|f_j^t,s_j^t)] + \beta\log\mathcal{N}(\mathbf{m}^t|\mathbf{0},\mathbf{K}^t+\mathbf{S}^t),$$

which is convenient from computational and programming point of view. Specifically, to compute this we need to perform a single Cholesky decomposition of $\mathbf{K}^t+\mathbf{S}^t$ which scales as $O((n^t)^3)$, i.e. cubically w.r.t. the size of the support set $n^t$. This is fine for small support sets (which is the standard case in few-shot learning) but it can become too expensive when $n^t$ becomes very large. However, given that the kernel has the linear form $k_\theta(x,x') = \phi(x;\theta)^\top\phi(x';\theta)$ (ignoring any kernel variance $\sigma_f^2$ for notational simplicity), where $\phi(x_i;\theta)$ is $M$-dimensional and given that $M \ll n^t$, we can also carry out the computations based on a Cholesky of a matrix of size $M \times M$. This is because $\mathbf{K}^t = \Phi^t{\Phi^t}^\top$, where $\Phi^t$ is an $n^t \times M$ matrix storing as rows the features vectors on the support inputs $X^t$, and therefore we can apply the standard matrix inversion and determinant lemmas for the matrix $\Phi^t{\Phi^t}^\top + \mathbf{S}^t$ when computing $\log\mathcal{N}(\mathbf{m}^t|\mathbf{0},\mathbf{K}^t+\mathbf{S}^t)$. Such $O(M^3)$ computations also gives us the quantities $q(f_j^v)$ and $q(f_j^t)$, as also explained next.

## C.3 DATA EFFICIENT GP META-TESTING PREDICTION WITH CONSTANT MEMORY

Once we have trained the GP meta learning system we can consider meta-testing where a new fresh task is provided having a support set $\mathcal{D}_*^t = (Y_*^t, X_*^t)$ based on which we predict at any arbitrary validation/test input $x_*$. This requires to compute quantities (such as the mean value $\mathbb{E}[y_*]$) associated with the predictive density

$$q(y_*) = \int p(y_*|f_*)p(f_*|\mathbf{f}_*^t,x_*,X_*^t)q(\mathbf{f}_*^t|\mathcal{D}_*^t)df_*d\mathbf{f}_*^t = \int p(y_*|f_*)q(f_*)df_*$$

where $q(f_*)$ is an univariate Gaussian given by

$$q(f_*) = \mathcal{N}(f_*|\mathbf{k}_*^t(\mathbf{K}^t+\mathbf{S}^t)^{-1}\mathbf{m}^t, k_* - \mathbf{k}_*^t(\mathbf{K}^t+\mathbf{S}^t)^{-1}{\mathbf{k}_*^t}^\top),$$

$$\mathbf{k}_*^t = \phi_*^\top\Phi^t, \ \mathbf{K}^t = \Phi^t{\Phi^t}^\top, \ k_* = \phi_*^\top\phi_*, \ \phi_* = \phi(x_*;\theta).$$

Here, $\Phi^t$ is an $n_*^t \times M$ matrix storing as rows the features vectors on the support inputs $X_*^t$. Note that if we wish to evaluate $q(y_*)$ at certain value of $y_*$, and given that the likelihood $p(y_*|f_*)$ is not the standard Gaussian, we can use 1-D Gaussian quadrature or Monte Carlo by sampling from $q(f_*)$.

An interesting property of the above predictive density is that when the support set $\mathcal{D}_*^t$ can grow incrementally, e.g. individual data points or mini-batches are added sequentially, the predictive density can be implemented with constant memory without requiring to explicit memorize the points in the support. The reason behind this that the feature parameters $\theta$ remain constant at meta-test time and the kernel function is linear, so we can apply standard tricks to update sufficient statistics as in Bayesian linear regression.

More precisely, what we need to show is that we can sequentially update the mean and variance of $q(f_*)$ with constant memory. $q(f_*)$ can be written as

$$q(f_*) = \mathcal{N}(f_*|\phi_*^\top{\Phi^t}^\top(\Phi^t{\Phi^t}^\top+\mathbf{S}^t)^{-1}\mathbf{m}^t, \phi_*^\top\left(I - {\Phi^t}^\top(\Phi^t{\Phi^t}^\top+\mathbf{S}^t)^{-1}\Phi^t\right)\phi_*)$$

$$= \mathcal{N}(f_*|\phi_*^\top({\Phi^t}^\top[\mathbf{S}^t]^{-1}\Phi^t+I)^{-1}{\Phi^t}^\top[\mathbf{S}^t]^{-1}\mathbf{m}^t, \phi_*^\top({\Phi^t}^\top[\mathbf{S}^t]^{-1}\Phi^t+I)^{-1}\phi_*) \quad (29)$$

where we applied the matrix inversion lemma backwards to write $I - {\Phi^t}^\top(\Phi^t{\Phi^t}^\top+\mathbf{S}^t)^{-1}\Phi^t = ({\Phi^t}^\top[\mathbf{S}^t]^{-1}\Phi^t+I)^{-1}$ and also used that ${\Phi^t}^\top(\Phi^t{\Phi^t}^\top+\mathbf{S}^t)^{-1} = {\Phi^t}^\top(\Phi^t{\Phi^t}^\top[\mathbf{S}^t]^{-1}+I)^{-1}[\mathbf{S}^t]^{-1} =$

$(\Phi^{t\top}[\mathbf{S}^t]^{-1}\Phi^t + I)^{-1}\Phi^{t\top}[\mathbf{S}^t]^{-1}$ (based on the identity $(AB+I)^{-1}A = A(BA+I)^{-1}$). Now observe that the $M$-dimensional vector $\mathbf{b}^t = \Phi^{t\top}[\mathbf{S}^t]^{-1}\mathbf{m}^t = \sum_{j=1}^{n^t}\phi(x_j^t;\theta)\frac{m_j^t}{s_j^t}$ can grow incrementally without memorizing the feature vectors $\phi(x_j^t;\theta)$ based on the recursion $\mathbf{b}^t \leftarrow \mathbf{b}^t + \phi(x_j^t;\theta)\frac{m_j^t}{s_j^t}$ (with the initialization $\mathbf{b}^t = 0$) as individual data points (similarly for mini-batches) are added in the support set: $\mathcal{D}^t \leftarrow \mathcal{D}^t \cup (x_j^t, y_j^t)$. Similarly, the $M \times M$ matrix $A^t = \Phi^{t\top}[\mathbf{S}^t]^{-1}\Phi^t = \sum_{j=1}^{n^t}\frac{1}{s_j^t}\phi(x_j^t;\theta)\phi(x_j^t;\theta)^\top$ can also be computed recursively with constant $O(M^2)$ memory.

Finally, note that the above constant memory during meta-testing can only be implemented when the feature vector $\theta$ remain fixed, which means that it is not applicable for the GP+MAML combination.

### C.4 MULTI-CLASS CLASSIFICATION

For multi-class classification meta learning problems we need to introduce as many latent functions as classes. For instance, when the number of classes for each task is $N$ we will need $N$ latent functions $f_n(x)$ which all are independent draws from the same GP. The marginal GP prior on the training and validation function values for a certain task factorizes as

$$\prod_{n=1}^{N} p(\mathbf{f}_n^v|\mathbf{f}_n^t, X^v, X^t)p(\mathbf{f}_n^t|X^t).$$

We assume a factorized encoding distribution of the form,

$$\prod_{n=1}^{N} p(\mathbf{f}_n^v|\mathbf{f}_n^t, X^v, X^t)q(\mathbf{f}_n^t|\mathcal{D}^t),$$

where each

$$q(\mathbf{f}_n^t|\mathcal{D}^t) = \mathcal{N}(\mathbf{f}_n^t|\mathbf{K}^t(\mathbf{K}^t + \mathbf{S}^t)^{-1}\mathbf{m}_n^t, \mathbf{K}^t - \mathbf{K}^t(\mathbf{K}^t + \mathbf{S}^t)^{-1}\mathbf{K}^t).$$

Here, $\mathbf{m}_n^t = Y_n^t \circ \widetilde{\mathbf{m}}^t$ and $Y_n^t$ is a vector obtaining the value 1 for each data point $x_j^t$ that belongs to class $n$ and $-1$ otherwise. Note that the encoding distributions share the covariance matrix and they only have different mean vectors. The representation of $\mathbf{m}_n^t$ makes the full encoding distribution permutation invariant to the values of the class labels. Since also we are using a shared (i.e. independent of class labels) amortized functions $\widetilde{m}_w(x)$ and $s_w(x)$ the terms $(\mathbf{S}^t, \widetilde{\mathbf{m}}^t)$ are common to all $N$ factors. This allows to compute the VIB objective very efficiently (in way that is fully scalable w.r.t. the number of classes $N$) by requiring only a single Cholesky decomposition of the matrix $\mathbf{K}^t + \mathbf{S}^t$. Specifically, by working similarly to C.2 we obtain the VIB objective per single task,

$$\sum_{j=1}^{n^v} \mathbb{E}_{q(\{f_{n,j}^v\}_{n=1}^N)}[\log p(y_j^v|\{f_{n,j}^v\}_{n=1}^N)] - \beta \sum_{n=1}^{N}\sum_{j=1}^{n^t} \mathbb{E}_{q(f_{n,j}^t)}[\log \mathcal{N}(m_{n,j}^t|f_{n,j}^t, s_j^t)]$$

$$+ \beta \sum_{n=1}^{N} \log \mathcal{N}(\mathbf{m}_n^t|\mathbf{0}, \mathbf{K}^t + \mathbf{S}^t),$$

where $q(\{f_{n,j}^v\}_{n=1}^N) = \prod_{n=1}^{N} q(f_{n,j}^v)$ and each univariate Gaussian $q(f_{n,j}^v)$ is given by the same expression as provided in C.3. The last two terms of the bound (i.e. the ones multiplied by the hyperparameter $\beta$) are clearly analytically computed, while the first term involves an expectation of a log softmax since the likelihood is

$$p(y_j^v = n|\{f_{n',j}^v\}_{n'=1}^N) = \frac{e^{f_{n,j}^v}}{\sum_{n'=1}^{N} e^{f_{n',j}^v}}.$$

To evaluate this expectation we apply first the reparametrization trick to move all tunable parameters of $q(\{f_{n,j}^v\}_{n=1}^N)$ inside the log-likelihood (so that we get a new expectation under a product of $N$ univariate standard normals) and then we apply Monte Carlo by drawing 200 samples.

Finally, note that to compute the predictive density we need to evaluate,

$$q(y_*) = \mathbb{E}_{q(\{f_{n,*}\}_{n=1}^N)}\left[p(y_*|\{f_{n,*}\}_{n=1}^N)\right],$$

which again is done by applying Monte Carlo by drawing 200 samples from $q(\{f_{n,*}\}_{n=1}^{N})$. This is precisely how the negative log-likelihood test performance was computed for the GP models in Table 3. To decide the classification label based on the maximum class predictive probability (in order to compute e.g. accuracy scores), we take advantage of the fact that all $N$ univariate predictive Gaussians $q(f_{n,*})$ have the same variance but different means, thus the predicted class can be equivalently obtained by taking the argmax of the means of these $N$ distributions.

### C.5 SPECIFIC GP IMPLEMENTATION AND AMORTIZATION FOR FEW-SHOT CLASSIFICATION

For all few-shot multi-class classification experiments in order to implement the GP and GP+MAML methods we need to specify the feature vector $\phi(x;\theta)$ and the amortized variational functions $\widetilde{m}_w(x)$ and $s_w(x)$. The feature vector is specified to have exactly the same neural architecture used in previous works for all datasets, Omniglot, mini-Imagenet and Augmented Omniglot; see for example Chen et al. (2019) for details on these architectures for all these three datasets. Note that when computing the GP kernel function the feature vector $\phi(x;\theta)$ is also augmented with the value 1 to automatically account for a bias term in the kernel function.

Regarding the two amortized variational functions needed to obtain the encoder, we consider a shared (with the GP functions) representation by adding two heads to the same feature vector $\phi(x;\theta)$: the first head corresponds to a linear output function $\widetilde{m}_w(x)$ and the second applies at the end the softplus activation $s_w(x) = \log(1 + \exp(a(x)))$ (since $s_w(x)$ represents variance) where the pre-activation $a(x)$ is obtained by a linear function of the feature vector. For numerical stability we also apply a final clipping by bounding these functions so that $\widetilde{m}_w(x) \in [-20, 20]$ and $s_w(x) \in [0.001, 20]$. The bound $-20$ and $20$ are almost never realized during optimization, so they are not so crucial, in contrast the lower bound $0.001$ on $s_w(x)$ is rather crucial regarding numerical stability since it ensures that the minimum eigenvalue of the matrix $\mathbf{K}^t + \mathbf{S}^t$ (i.e. the matrix we need to decompose using Cholesky) is bounded below by $0.001$.

## D  FURTHER EXPERIMENTAL DETAILS AND RESULTS

### D.1 EXPERIMENTAL SETTINGS AND HYPERPARAMETERS

The memory-based GP method (referred to in the Tables as GP) is trained by the VIB objective, as defined in Section 3, where the kernel feature vector $\phi(x;\theta)$ is obtained by the last hidden layer of the same neural architecture as used in MAML. Based on this $M$-dimensional feature vector $\phi(x;\theta) \in \mathbb{R}^M$ we consider two kernel functions: the standard linear kernel function $k_\theta(x,x') = \frac{1}{M}\phi(x;\theta)^\top \phi(x';\theta)$ (where the kernel variance $\sigma_f^2$ is fixed to $1/M$) and the cosine kernel $k_\theta(x,x') = \frac{\phi(x;\theta)^\top \phi(x';\theta)}{||\phi(x;\theta)||||\phi(x';\theta)||}$. These kernels are used in the experiments.

The Omniglot dataset consists of 20 instances of 1623 characters from 50 different alphabets. Each instance was drawn by a different person. It is augmented by creating new characters which are rotations of each of the existing characters by 0, 90, 180 or 270 degrees. In Omniglot experiment, MAML, Stochastic MAML and GP+MAML run by applying one adaptation step for both meta-training and meta-testing. The mini-Imagenet involves 64 training classes, 12 validation and 24 tests classes. Following previous work on mini-Imagenet we meta-train Stochastic MAML with 5 adaptation steps, while 10 steps are used for meta-testing. GP+MAML uses 5 steps for both meta-training and meta-testing. For both Omniglot and mini-Imagenet we follow the experimental protocol proposed by Finn et al. (2017), which involves fast learning of $N = 5$-way classification with $K = 1$ or $K = 5$ shots. The problem of $N$-way classification is set up as follows: select $N$ unseen classes, provide the model with $K$ different instances per class, and evaluate the model's ability to classify new instances within the $N$ classes.

The Augmented Omniglot benchmark is a modified version of Omniglot which necessitates long-horizon adaptation and it is often considered as many-shots problem (Chen et al., 2019). For each alphabet, 20 characters are sampled to define a $N = 20$-class classification problem with $K = 15$ shots. Further, both train and validation images are randomly augmented, by applying transformations, which makes it even more challenging. Following Flennerhag et al. (2018); Chen et al. (2019), during meta-testing both MAML and Stochastic MAML perform 100 steps of adaptation (resulting in 2000 data points seen in total by the model where each step processes a minibatch of size 20

points), while they are meta-trained by applying 20 adaptation steps (i.e. 400 training points seen per task). Both GP methods are meta-trained by memorizing the full $N \times K = 20 \times 15 = 300$ support points without further data augmentation, while during meta-testing we allow the GP methods to see up to 2000 points. For all methods we perform a hyperparameter search using the train and validation subsets of all three benchmarks as detailed below.

For sinusoid regression, Omniglot and mini-Imagenet, meta-training of all methods consisted of 60000 iterations or episodes, where in each episode a learning update is performed based on a mini-batch of tasks. For the sinusoid regression the meta batch-size was 5 tasks and for each task we have $K = 10$ examples. For $N$-way, $K$-shot classification in Omniglot and mini-Imagenet (with $N = 5$, and $K = 1, 5$ as mentioned in Table 2 in the main paper), the meta batch was 32 tasks in Omniglot and 4 tasks in mini-Imagenet. Also for these two datasets Stochastic MAML uses one gradient step (in the inner loop) for both meta-training and meta-testing in Omniglot, while in mini-Imagenet it uses 5 and 10 steps respectively, i.e. exactly as MAML by Finn et al. (2017) was applied in these datasets. As mentioned in the main paper GP+MAML considers one gradient adaptation step in Omniglot and 5 in mini-Imagenet (for both meta-training and meta-testing). The neural architectures of all these experiments is the same as in Finn et al. (2017). Specifically, for the Omniglot the architecture is from Vinyals et al. (2016), which has 4 modules with a $3 \times 3$ convolutions and 64 filters, followed by batch normalization, a ReLU nonlinearity, and $2 \times 2$ max-pooling. The Omniglot images are downsampled to $28 \times 28$, so the dimensionality of the last hidden layer is 64. For Omniglot, strided convolutions are used instead of max-pooling. For the GP methods this constructs 65-dimensional feature vector $\phi(x; \theta)$ (64 plus one for the bias term which is included in $\phi(x; \theta)$). For mini-Imagenet, the network uses 32 filters per layer and the final layer the feature vector feature is obtained by flattening so that finally the feature vector $\phi(x; \theta)$ is 801-dimensional.

For Augmented Omniglot, the meta-training of all methods consisted of 5000 iterations. The Augmented Omniglot dataset is a modified version of Omniglot which, for gradient-based methods like MAML, necessitates long-horizon adaptation and it is often considered as many-shots problem (Chen et al., 2019). For each alphabet, 20 characters are sampled to define a 20-class classification problem with $K = 15$ data points per class. Furthermore, both train and test images are randomly augmented, by applying transformations. Following Flennerhag et al. (2018); Chen et al. (2019), we use a 4-layer convnet and during meta-testing MAML and Stochastic MAML performs 100 steps of adaptation (resulting in 2000 data points where each step processes a minibatch of size 20 data-points, i.e. per class), while they are meta-trained by applying 20 adaptation steps. GP+MAML uses 5 adaptation steps for both meta training and meta testing.

Additional hyperparameters details are included in Tables 4, 5 and 6. The ranges for the hyperparameters' search are:

- Outer learning rate $\alpha$:

$$[0.0001, 0.00025, 0.0005, 0.00075, 0.001, 0.002, 0.0025,$$
$$0.005, 0.0075, 0.01, 0.025, 0.05, 0.075, 0.1]$$

- Inner learning rate $\rho$:

$$[0.0001, 0.0005, 0.001, 0.002, 0.005, 0.01, 0.025, 0.05, 0.1, 0.2, 0.5, 1.0, 1.5, 2.0]$$

- Bottleneck coefficient $\beta$:

$$[1e-07, 1e-06, 1e-05, 5e-05, 0.0001, 0.00025, 0.0005,$$
$$0.00075, 0.001, 0.0025, 0.005, 0.0075, 0.01, 0.1, 0.5, 1.0]$$

## D.2 ADDITIONAL EXPERIMENTAL RESULTS

Table 7 provides detailed few-shot sinusoid regression results, while Figure 3 illustrates how the GP model, after having been meta-trained on the sinusoid regression tasks, predicts a test task as the number $K$ of the training examples increases.

We also found that the qualitative behaviour of GP and GP+MAML are quite different. In Figure 4, we report the performance of the GP and GP+MAML on mini-Imagenet tasks as a function of

| Parameter | Stochastic MAML | |
|---|---|---|
| | **Omniglot** | |
| | $K = 1$ | $K = 5$ |
| Outer l.r. $\alpha$ | 0.005 | 0.001 |
| Inner l.r. $\rho$ | 1.5 | 1.5 |
| Bottleneck $\beta$ | $10^{-7}$ | $10^{-6}$ |
| | **mini-Imagenet** | |
| | $K = 1$ | $K = 5$ |
| Outer l.r. $\alpha$ | 0.0005 | 0.0005 |
| Inner l.r. $\rho$ | 0.05 | 0.1 |
| Bottleneck $\beta$ | $10^{-5}$ | $10^{-4}$ |
| | **Augmented Omniglot** | |
| Outer l.r. $\alpha$ | 0.01 | |
| Inner l.r. $\rho$ | 0.5 | |
| Bottleneck $\beta$ | $10^{-6}$ | |
| | **Sinusoid regression** | |
| Outer l.r. $\alpha$ | 0.0005 | |
| Inner l.r. $\rho$ | 0.1 | |
| Bottleneck $\beta$ | $10^{-3}$ | |

**Table 4:** Hyperparameters for Stochastic MAML.

| Parameter | MAML |
|---|---|
| | **Augmented Omniglot** |
| Outer l.r. $\alpha$ | 0.01 |
| Inner l.r. $\rho$ | 0.5 |
| | **Sinusoid regression** |
| Outer l.r. $\alpha$ | 0.002 |
| Inner l.r. $\rho$ | 0.005 |

**Table 5:** Hyperparameters for MAML in sinusoid regression and Augmented Omniglot.

| Parameter | GP (cos) | | GP (linear) | | GP + MAML (cos) | | GP + MAML (linear) | |
|---|---|---|---|---|---|---|---|---|
| | **Omniglot** | | | | | | | |
| | $K=1$ | $K=5$ | $K=1$ | $K=5$ | $K=1$ | $K=5$ | $K=1$ | $K=5$ |
| Outer l.r. $\alpha$ | 0.001 | 0.0025 | 0.0025 | 0.005 | 0.00075 | 0.005 | 0.0025 | 0.001 |
| Inner l.r. $\rho$ | − | − | − | − | 0.0005 | 0.002 | 0.002 | 0.0005 |
| Bottleneck $\beta$ | 0.0005 | 0.0005 | 0.0001 | 0.0001 | 0.0005 | 0.0005 | 0.0001 | 0.00005 |
| | **mini-Imagenet** | | | | | | | |
| | $K=1$ | $K=5$ | $K=1$ | $K=5$ | $K=1$ | $K=5$ | $K=1$ | $K=5$ |
| Outer l.r. $\alpha$ | 0.00075 | 0.0005 | 0.0025 | 0.00075 | 0.001 | 0.0005 | 0.001 | 0.00075 |
| Inner l.r. $\rho$ | − | − | − | − | 0.0005 | 0.0005 | 0.002 | 0.002 |
| Bottleneck $\beta$ | 0.005 | 0.005 | 0.005 | 0.005 | 0.001 | 0.0005 | 0.0001 | 0.0005 |
| | **Augmented Omniglot** | | | | | | | |
| Outer l.r. $\alpha$ | 0.0025 | | 0.05 | | 0.005 | | 0.005 | |
| Inner l.r. $\rho$ | − | | − | | 0.001 | | 0.0005 | |
| Bottleneck $\beta$ | 0.0005 | | 0.00075 | | 0.00025 | | 0.00001 | |
| | **Sinusoid regression** | | | | | | | |
| Outer l.r. $\alpha$ | 0.00075 | | − | | 0.002 | | − | |
| Inner l.r. $\rho$ | − | | − | | 0.0001 | | − | |
| Bottleneck $\beta$ | 1.0 | | − | | 0.5 | | − | |

**Table 6:** Hyperparameters for the GP methods.

| Model | K=5 | K=10 | K=20 |
|---|---|---|---|
| MAML (our) | | | |
| grads steps 1 | 0.600±0.662 | 0.359±0.015 | 0.228±0.018 |
| grads steps 5 | 0.311±0.013 | 0.12±0.006 | 0.06±0.004 |
| grads steps 10 | 0.280±0.013 | 0.096±0.005 | 0.043±0.003 |
| Stochastic MAML | | | |
| grads steps 1 | 0.662±0.04 | 0.382±0.021 | 0.244±0.014 |
| grads steps 5 | 0.352±0.035 | 0.141±0.014 | 0.073±0.006 |
| grads steps 10 | 0.317±0.34 | 0.116±0.012 | 0.054±0.004 |
| GP | 0.02±0.014 | 0.002±0.001 | 0.001±0.001 |
| GP+MAML | 0.058±0.054 | 0.002±0.001 | 0.002±0.002 |
| MAML (Finn et al., 2017) | 0.35 | − | − |

**Table 7:** Detailed few-shot sinusoid regression results. We report the results of the GP model for $K = 5, 10, 20$ and for MAML assuming different gradient adaptation steps (including also a result reported in the original MAML paper by Finn et al. (2017)).

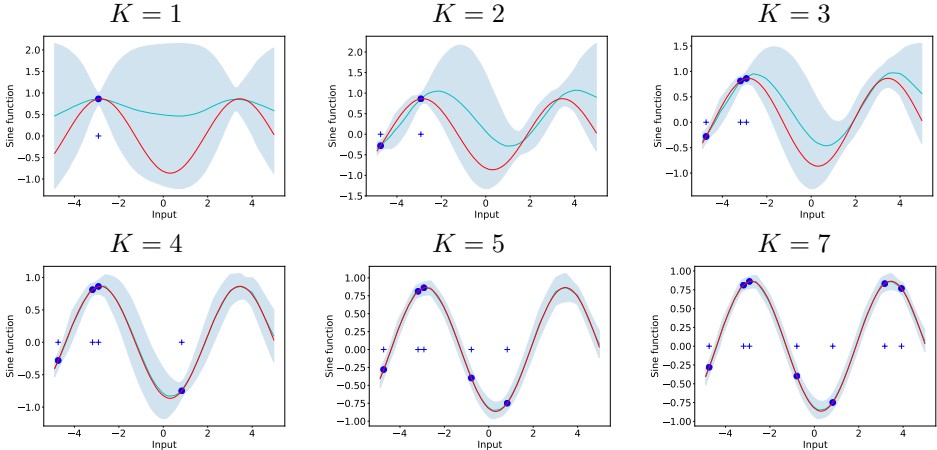

**Figure 3:** The red curve represents a ground truth sinusoid function whereas the light blue one is the mean prediction of a GP. As $K$ increases, the uncertainty (shown by the shaded area) of the GP decreases and its mean predictions become closer to the ground truth. Given already $K = 4$ points, the mean prediction matches well the sinusoid function.

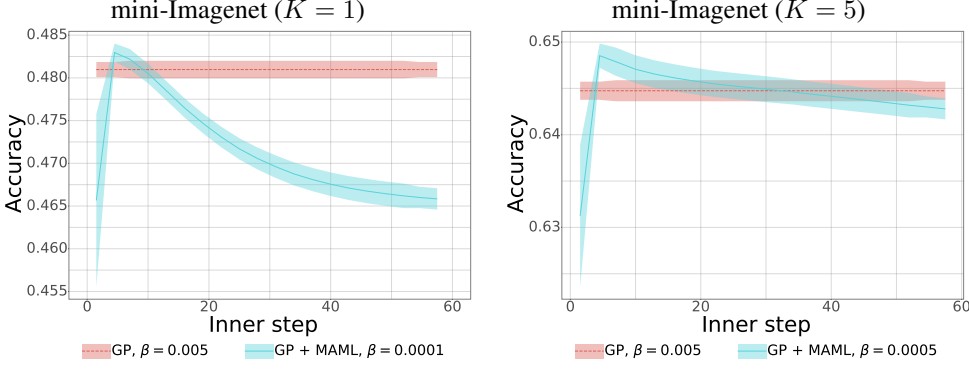

**Figure 4:** Qualitative difference between GP and GP+MAML on mini-Imagenet as a function of inner loop steps. Because there is no inner loop for the GP, we simply report it as a reference to the GP+MAML.

inner steps executed at test time by GP+MAML. Note, that GP actually has only one step executed over all the test-time data in once, because there is no inner loop for it. Intuitively, we would expect GP+MAML to have similar performance to GP at the beginning of inner loop, but this is not what is happening. Instead, it starts quite low and peaks at exactly the number of inner steps used during meta-training and then the performance starts to deteriorate.

## D.3 ABLATIVE STUDIES

**Bottleneck cost $\beta$ ablation** To understand the impact of $\beta$ on learning we provide the ablation analysis in Figure 5 on sinusoid regression and two few-shot classification tasks. For a regression task, Figure 5 (a), we observe that having a large $\beta = 1.0$ is beneficial and provides the best mean squared error (MSE). For few-shot classification tasks Figures 5 (b)-(f), we observe that there is a large range of $\beta$ values in between $[10^{-7}, 0.01]$ providing similar classification accuracy. Interestingly, for values of $\beta$ close to 1, where for $\beta = 1$ the VIB corresponds to an ELBO, the performance deteriorates significantly.

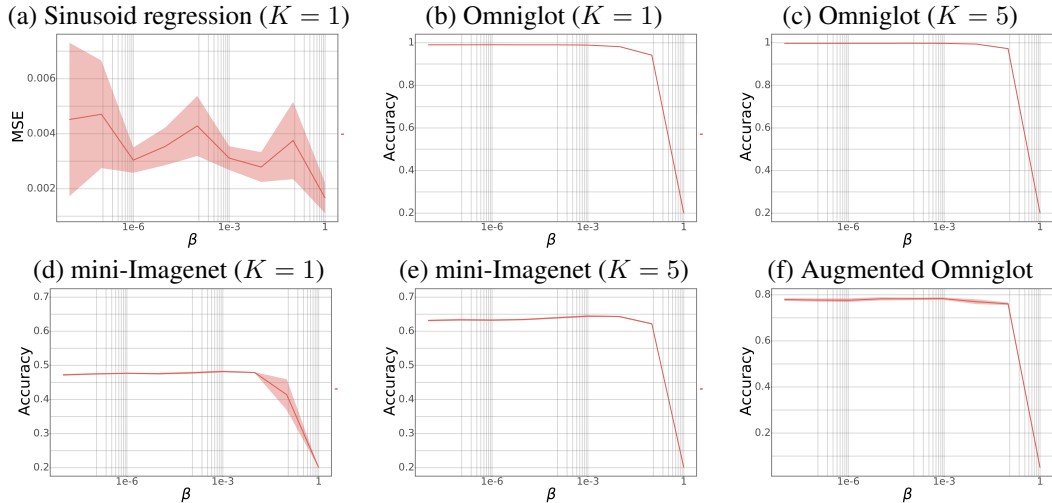

**Figure 5:** Ablation analysis for different $\beta$ values on sinusoid regression and few shot classification tasks. For sinsusoid regression we report mean squared error (MSE): lower is better. For few-shot classification tasks, we report the accuracy: higher is better.

## D.4 IMPROVED VERSIONS FOR THE GP-BASED METHODS

For GP-based methods we consider additional popular tricks used in the community to improve performance.

For Augmented Omniglot dataset (Flennerhag et al., 2018), for the network architecture we add 4 additional layers similar to Flennerhag et al. (2020), where each layer is added after a convolutional layer. These additional layers are simple convolutions as considered in Flennerhag et al. (2020). In addition, we add a batch normalization. Simply using more layers in GP methods significantly improves the performance on this dataset. Note, that the results are achieved without a special warp-grad architecture as in Flennerhag et al. (2020). In Table 8 we show the results with and without batch normalization (BN).

In addition, we run additional experiments on mini-ImageNet (Ravi & Larochelle, 2017) and tiered-ImageNet (Ren et al., 2018) with using pre-trained embeddings from LEO (Rusu et al., 2019) paper as features. The embeddings are taken from their GitHub repository. On top of these features, we construct an MLP with two hidden layers and ReLU activations where each layer has 128 hidden dimensions. This defines the feature vector $\phi(x; \theta)$ which is used by the GP methods in all the experiments. Moreover, we apply dropout to the LEO embeddings before passing them to the MLP. The results are given in Table 9. The hyperparameters are tuned in the same way as for the previous experiments. Their parameters are given in Table 10. Notice that the GP methods are comparable

|  | **Augmented Omniglot** 20-way |
|---|---|
| **Model** | |
| GP (linear) | 86.29±0.84% |
| GP (cos) | 85.6±0.76% |
| GP (linear), BN | 86.62±1.3% |
| GP (cos), BN | ***86.7**±0.9% |
| GP+MAML (linear) | 84.59±0.5% |
| GP+MAML (cos) | 84.23±1.0% |
| GP+MAML (linear), BN | 85.94±0.7% |
| GP+MAML (cos), BN | ***86.17**±1.04% |
| Warp-Leap Flennerhag et al. (2020) | 83.6±1.9% |

**Table 8:** Classification test accuracy performance on Augmented Omniglot for improved architectures. The GP based methods use 4 additional simple convolutional layers where each of these layers is added after each network layer, similar to Flennerhag et al. (2020). We also report results with and without batch normalization.

|  | **mini-ImageNet** 5-way | | **tiered-ImageNet** 5-way | |
|---|---|---|---|---|
| **Model** | $K = 1$ | $K = 5$ | $K = 1$ | $K = 5$ |
| GP (linear) | 60.4±0.28% | 77.1±0.08% | 64.16±0.16% | 81.6±0.09% |
| GP (cos) | 60.8±0.16% | 76.8±0.08% | 65.3±0.16% | 81.3±0.22% |
| GP + MAML (linear) | 60.5±0.5% | 77.08±0.13% | 64.5±0.22% | **81.6±0.16**% |
| GP + MAML (cos) | 60.8±0.19% | 77.1±0.21% | 65.6±0.2% | 81.5±0.14% |
| LEO Rusu et al. (2019) | **61.76±0.08**% | **77.59±0.12**% | **66.33±0.05**% | 81.44±0.09% |

**Table 9:** Additional results on mini-ImageNet and tiered-ImageNet. We re-use LEO Rusu et al. (2019) embeddings from their GitHub code and apply a 2-layers MLP with ReLU activations and 128 hidden dimensions to construct the feature vector.

with LEO, with no significant difference. We would like to point out that LEO uses a set of different tricks: dropout, label smoothing, $\ell_2$ regularization and orthogonality penalty (Rusu et al., 2019), while we have only considered dropout.

|  | **GP** (cos) | | **GP** (linear) | | **GP+MAML** (cos) | | **GP+MAML** (linear) | |
|---|---|---|---|---|---|---|---|---|
|  | $K = 1$ | $K = 5$ | $K = 1$ | $K = 5$ | $K = 1$ | $K = 5$ | $K = 1$ | $K = 5$ |
| **mini-ImageNet** | | | | | | | | |
| Outer l.r. $\alpha$ | $5 * 10^{-5}$ | $2.5 * 10^{-5}$ | 0.00025 | $7.5 * 10^{-5}$ | $5 * 10^{-5}$ | $7.5 * 10^{-5}$ | $2.5 * 10^{-5}$ | 0.0001 |
| Inner l.r. $\rho$ | – | – | – | – | 0.001 | 0.001 | 0.001 | 0.001 |
| Bottleneck $\beta$ | 0.0025 | $10^{-6}$ | 0.00075 | $10^{-6}$ | 0.0001 | 0.0001 | 0.0001 | $10^{-6}$ |
| Dropout $p$ | 0.1 | 0.1 | 0.1 | 0.2 | 0.1 | 0.2 | 0.1 | 0.2 |
| **tiered-ImagetNet** | | | | | | | | |
| Outer l.r. $\alpha$ | $7.5 * 10^{-5}$ | 0.0001 | $7.5 * 10^{-5}$ | 0.00025 | $7.5 * 10^{-5}$ | $7.5 * 10^{-5}$ | 0.0001 | 0.0001 |
| Inner l.r. $\rho$ | – | – | – | – | 0.001 | 0.001 | 0.001 | 0.001 |
| Bottleneck $\beta$ | $10^{-5}$ | $10^{-5}$ | $5 * 10^{-7}$ | $10^{-6}$ | $10^{-8}$ | $10^{-8}$ | $10^{-7}$ | $10^{-7}$ |
| Dropout $p$ | 0.2 | 0.1 | 0.1 | 0.2 | 0.2 | 0.1 | 0.2 | 0.2 |

**Table 10:** Hyperparameters for the additional results on mini-ImageNet and tiered-ImageNet. For dropout rate, we considered values: 0.0, 0.1, 0.2, 0.3, 0.4, 0.5, 0.6, 0.7.

