# OpenReview forum: "Information Theoretic Meta Learning with Gaussian Processes"
_ICLR.cc/2021/Conference — Reject_

### Official Review · AnonReviewer1 · 2020-10-26
**Review of Information Theoretic Meta Learning with Gaussian Processes:  Interesting idea. Experimental part is weak.**

**Rating:** 5
**Confidence:** 4

**Review:**

## Summary

The paper derives a meta-learning framework based on the information bottleneck principle. By adapting the variational approximation proposed in [1] to the meta-learning setting, the authors come up with a tractable objective that generalises both gradient based and memory meta-learning methods. Based on this framework, the authors proposed a new memory based meta-learning algorithm by using a GP with a deep kernel and an extension that combines this memory based method with MAML. The authors show that the method outperforms MAML in several standard meta-learning  tasks, especially in regression and many shots classification problems.

## Comments

The motivation of the paper is really clear and the paper is extremely well-written.

From the technical perspective, the paper has two main contributions:

1. The development of a general meta-learning framework based on the information bottleneck that generalizes gradient and memory based meta-Learning
2. Two new methods that resort to GPs with deep kernels to implement the encoder in the aforementioned framework

Regarding the framework, the extension of the variational information bottleneck to the meta-learning setting is not particularly challenging being a straightforward extension of [1]. However, it is an interesting observation that this extension subsumes gradient based methods (by using parametric encodings) and memory based ones (by using nonparametric encodings). One of the things that it is missing is that the authors failed to mention that the connection between meta-learning and the information bottleneck principle had been previously explored in the literature (see [2]). While the setting is slightly different, it would make sense for the authors to address how the two papers are related, especially since the GP method proposed by the authors can be used in the transductive settings as well.

Regarding the new methods, I think that the use of a GP encoder with a deep kernel is interesting but again not particularly challenging from the technical perspective. However, I think the combination of MAML + GP it is a very interesting observation that can open new research avenues.

The weakest part of the paper is the experiments. The authors use MAML as the unique baseline ignoring more up to date meta-learning methods which undermines the message of the paper. It would be really interesting to see how the method generalises and compares against other state of the art methods. For example, there has been several methods that try to leverage uncertainty into MAML (e.g. [3]). This would be an interesting comparison since it would help to disentangle the impact of a parametric/nonparametric encoder versus probabilistic/deterministic encoders. In addition, there are several recent methods specifically developed for transductive meta-learning that would be interesting to include in the comparison (e.g. [4]). Finally, I would appreciate a more in depth analysis of why the combined approach works better or worse than the GP one in different settings (see some questions below).

## Questions/Minors

* In section 2.2 the authors cast MAML under their framework. Two do so, they assume Z_i=\psi_i leading to p_\theta(\mathcal{D}_i^v|\psi_i) and p_\theta(\psi_i). However, the second approximation is irrelevant since further on they assume \beta=0.
* To build the stochastic extension of MAML, the authors use a stochastic encoder. However, in addition they use \beta \neq 0. It would be nice to see an ablation study to see the contribution of these separate extensions (there is an ablation study in D.3 but it is not specifically addressing this and it is not clear what is the model they use for the study). Also it would be interesting a small discussion about the importance of tying the variances of the prior and the encoder
* In section 3.1 it should be mention that it is a degenerate GP.
* Toward the end of section 3.2 a bunch of simplifications are done in order to come up with the final version of the combined algorithm: \beta to \beta_f and \beta_\psi and \beta_psi=0 which leads to a deterministic encoder and remove the need of defining a prior. However, these simplifications goes in the opposite direction of the extensions made to come up with the stochastic version of MAML. It would be great if the authors could elaborate on why that is the case and the contribution to the final performance of each of this simplifications.

## References

[1] Deep variational information bottleneck
[2] Empirical Bayes Transductive Meta-Learning with Synthetic Gradients
[3] Recasting Gradient-Based Meta-Learning as Hierarchical Bayes
[4] Learning to Propagate Labels: Transductive Propagation Network for Few-shot Learning

---

> ### Author Response · Authors · 2020-11-18
> **response to the main comments**
>
> "One of the things that it is missing is that the authors failed to mention that the connection between meta-learning and the information bottleneck principle had been previously explored in the literature (see [2]). While the setting is slightly different, it would make sense for the authors to address how the two papers are related, especially since the GP method proposed by the authors can be used in the transductive settings as well."
>
> Thank you for this comment and for pointing to this previous work that we were not aware of.  We have added the following paragraph to the related work section that we believe it describes accurately the connection with this previous work:
> *A meta learning method that connects with the information bottleneck was recently proposed by Hu et al. (2020). There the information bottleneck was used to analyze the generalization of a variational Bayesian inference objective suitable for transductive supervised few-shot learning. Note that the information bottleneck derived  in Hu et al. (2020) (theorem 1) is not the same as the information bottleneck objective used here (i.e.\ the objective in 7 for the supervised learning case), since they differ in the second term. In addition, our framework  expresses a general information bottleneck principle for meta learning applicable beyond transductive supervised learning.*
>
> "The weakest part of the paper is the experiments. The authors use MAML as the unique baseline ignoring more up to date meta-learning methods which undermines the message of the paper. It would be really interesting to see how the method generalises and compares against other state of the art methods. For example, there has been several methods that try to leverage uncertainty into MAML (e.g. [3]). This would be an interesting comparison since it would help to disentangle the impact of a parametric/nonparametric encoder versus probabilistic/deterministic encoders. In addition, there are several recent methods specifically developed for transductive meta-learning that would be interesting to include in the comparison (e.g. [4]). Finally, I would appreciate a more in depth analysis of why the combined approach works better or worse than the GP one in different settings (see some questions below)."
>
> Then main reason we focus on MAML is that it is still very competitive in these few-shot learning and many other methods build on MAML.
> We will try to reproduce and re-run one of the probabilistic or Bayesian MAML methods based on available code. Note that while our framework can explain transductive settings, our current experiments do not use any specialised transductive procedures (as those in [2] or [4]). For instance, if you see the GP predictive posterior $q(f_{i,j}^v)$ (below equation 11) essentially implies a non- transductive setting, since this posterior depends on the union of the single validation input and the support set. The only transductive component of the method comes from the batch-normalisation which is part of the neural architecture of Finn et al (2017) that we use in all experiments without any modification. To build a more specialised transductive procedure we will need to significantly change the way we currently set the encoder (i.e. the way we amortise the GP; see Appendix C.1), and we feel this is beyond the scope of this paper.
>
> Also note that in the currently revised version we have included  additional experiments that show that our GP method can be significantly more data efficient in meta-testing than gradient-based methods such as MAML, with better calibrated probabilities.  This is shown in Fig2 and described in the last paragraph in the experiments section.
>
> "In section 2.2 the authors cast MAML under their framework..."
>
> We agree this is relevant for stochastic MAML method where $\beta>0$. We can clarify this.
>
> "To build the stochastic extension of MAML, the authors use a stochastic encoder...... (there is an ablation study in D.3 but it is not specifically addressing this and it is not clear what is the model they use for the study). Also it would be interesting a small discussion about the importance of tying the variances of the prior and the encoder"
>
> The model we use for $\beta$ ablation is just the GP, i.e. the only memory-based system. For GP+MAML the ablation study
> for $\beta$ shows a very similar picture with the GP. We haven't done any ablation study for Stochastic MAML but we plan to follow
> reviewer's suggestion to study the effect of tying the variances. as well the effect of $\beta_\psi =0$ vs $\beta_\psi>0$ in GP+MAML.

---

### Official Review · AnonReviewer4 · 2020-10-27
**Interesting ways of combining information theory, variational inference, and Bayesian non-parametric for meta-learning.**

**Rating:** 5
**Confidence:** 3

**Review:**

Summary:

The paper proposed variational approximations to the information bottleneck objective functions for meta-learning.
The authors then provided three different settings using their variational loss functions, namely SMAML, GP, and GP + MAML.
The authors' motivations for these three settings were to study the effect of stochastic gradient-based method, non-parametric method, and the combination of gradient-based method with non-parametric method.

Reason for the score:

I find the paper interesting for establishing a variational information theoretic objective function for meta-learning.
I also enjoyed the comparisons among purely gradient-based method with non-parametric method and their hybrid.
Yet, I did find some details where the motivations are a bit difficult to understand. It would be great if the authors could
clarify them.

Pro:
- The authors provided clear derivations for their variational objective functions.
- The authors provided clear kernel setups for their GP models.
- The authors provided evaluations for their three settings on both regression and classification tasks.
- The paper is easy to follow and I did not find typos.

Con / Questions:
- For the objective function in Eq.1, is there a particular reason for assuming conditional independence for $q(Z| D^{t}) $?
- For Eq.5 and Eq.6, are the models of $p_\theta (D_i^v | Z_i)$ and $p_\theta (Z_i)$ related in some ways? It seems that they are sharing the parameters $\theta$.
- I did not find Section 3.2's motivation very convincing. The authors mentioned that the hope for combining MAML with GP is to reduce inner steps for MAML and possibly achieve better results. Yet, in Figure 4, it seems that pure GP is basically on-par with GP+MAML and outperforms GP+MAML when more inner steps are introduced for MAML. Could the authors help me with some insights for this observation?
- It seems that SMAML performed worse than MAML consistently for regression tasks and occasionally better for classification tasks. I observe that SMAML differ from MAML in two aspects, being stochastic and having the extra KL regularizer controlled by $\beta$. Are the worsened performance due to being stochastic or too much regularization from the KL term?
- On the classification tasks, it seems that GP models perform better than the standard MAML mostly when $K$ is large. It is difficult to conclude that the results are better thanks to GP being non-parametric or the variational loss function being better. This is also somewhat true for the regression tasks because we know GP is smooth and interpolates better.

-----------------------------------------------------------------
Post Rebuttal:

Many thanks for the authors to update their original paper addressing some of my questions and concerns.
Unfortunately, I still think that some aspects could be better analysed,
it is not crystal clear to me if the improvements come from the GP or their proposed variational information
theoretic function. I am keeping my original score.

---

> ### Author Response · Authors · 2020-11-18
> **response to the main points**
>
>
> "For the objective function in Eq.1, is there a particular reason for assuming conditional independence.."
>
> The main motivation of $q(Z|D^t)$ is that at test time you will only be able to observe the training set $D^t$ and be asked to predict $D^v$. In general, the rationale behind  building the encoder $q(Z| \cdot)$ is that you would like to condition only on things that you will observe at test time. For example, in supervised meta learning at test time you will observe a task labelled training set $D^t = (y^t, X^t)$ but also unlabelled validation/test inputs $X^v$, so in such case $q(Z | y^t, X^t, X^v)$.
>
> "For Eq.5 and Eq.6, are the models of  and  related in some ways? It seems that they are sharing the parameters"
>
> We have made this choice mainly for notational simplicity. Since both $p(Z)$ and $p(D^v |  Z)$ are part of the decoding/generative process, as opposed to the encoding  or recognition model $q(Z|D^t)$ that goes in reverse, they could naturally share some tunable parameters (although this is not necessary). Note that as we mention in footnote 1, all derivations are valid irrespectively of whether $\theta$ and $w$ have shared components or not.
>
> "I did not find Section 3.2's motivation very convincing. The authors mentioned that the hope for combining MAML with GP is to reduce inner steps for MAML and possibly achieve better results. Yet, in Figure 4, it seems that pure GP is basically on-par with GP+MAML and outperforms GP+MAML when more inner steps are introduced for MAML. Could the authors help me with some insights for this observation?"
>
> Our main motivation of GP+MAML is to be able to show that based on the information bottleneck we are able to combine  memory-based and gradient-based meta learning. The reason of the behaviour in Fig4 is that GP+MAML, due its MAML component, is optimised during meta-training to reconstruct well the validation set (so essentially to have good performance on the test data) after performing a fixed number of gradients steps for the feature vector $\phi(x;\theta)$ using the support data. So somehow the system is meta-trained to provide the best performance not with the initial $\theta$ but with the $\psi_i$ obtained from $\theta$ after applying a **fixed** number of SGD steps. After this number of steps the performance can deteriorate, simply because the $\psi_i$ can start overfitting due to the fact that  for this new task we have very few data (say only N=5,K=1 in mini-Imagenet) and we keep cycling with SGD updates over it.  Overfitting will not occur if for some reason for the new task we have much more data, so the performance of GP+MAML in such case could keep improving. We believe what we observe in Fig4 can be a typical behaviour of any MAML method.
>
> "It seems that SMAML performed worse than MAML consistently for regression tasks and occasionally better for classification tasks. I observe that SMAML differ from MAML in two aspects, .....GP is smooth and interpolates better."
>
> We have included some additional experiments that show that our GP method can be significantly more data efficient than gradient-based methods such as MAML, with better calibrated probabilities. Note also that the information bottleneck plays a crucial role, especially via the value of $\beta$, in the GP performance. In Fig5 in the Appendix we include an ablation study for $\beta$ that shows that small values for $\beta$ for the classification problems are preferred. Values of $\beta$ close to $1$ (that will make training more like Bayesian variational inference) provide the worse performance.

---

### Official Review · AnonReviewer2 · 2020-10-28

**Rating:** 4
**Confidence:** 3

**Review:**

This paper formulates meta-learning using the information bottleneck. The method develops a meta-learning algorithm based on Gaussian Processes, which they also interpret as a memory-based algorithm. They further describe an extension that combines the GP-based method and MAML.

The paper seems to lack focus. It consists of two orthogonal components: (1) applying the information bottleneck to meta-learning and (2) a GP-based meta-learning algorithm. Each of these components is a separate meta-learning algorithm: (1) alone can extend MAML to "stochastic MAML" ($\beta > 0$) and (2) alone is "GP" with hyperparameter $\beta=0$. The paper does not make a sufficient case for why these two components should be used in tandem (e.g. GP w/ $\beta > 0$), nor does it perform ablation experiments that each positively contributes to some measure of meta-learning performance.

The interpretation of MAML as a special case of information bottleneck (section 2.2) is not particularly novel: as the paper mentions in section 4, any previous works (LLAMA, probabilistic MAML, bayesian MAML, iMAML) have built on the interpretation of the initial parameter of MAML as encoding a prior distribution over parameters, and the gradient step as posterior inference. The fact that information bottleneck with $\beta=0$ reduces to standard learning is well-known.

The experimental results are mixed, and the paper doesn't give much of an interpretation of these results. GP and GP+MAML outperform MAML by an order of magnitude on the sinusoid regression problem but is outperformed by MAML in two of the five classification tasks considered. The experiments in the appendix are similarly mixed. GP wins on Augmented Omniglot but loses on ImageNet variants. Is the GP approach vastly superior on only regression problems, or is the difference just due to tuning and the fact that sinusoid regression is a toy problem?

Given the relation to other probabilistic extensions of MAML, perhaps comparing against at least one of them would have made experiments more informative.

The experiments about the efficiency regarding the amount of data (Figure 2 right) are promising, but this claim would be more significant if evaluated on multiple tasks.

minor

There is a link formatting issue btw pages 2 and 3

---

> ### Author Response · Authors · 2020-11-18
> **response to the main points**
>
> "The paper seems to lack focus. It consists of two orthogonal components: (1) applying the information bottleneck to meta-learning and (2) a GP-based meta-learning algorithm. ..."
>
> The main objective of our paper is to derive  a general information bottleneck principle for meta learning. We believe this is a valid methodological contribution. Then to justify this as a valid principle we have fully applied it to GPs. We believe that information bottleneck allows to connect and unify memory-based and gradient-based meta learning under the concept of "encoding in the bottleneck", so that memory-based methods use parametric encodings while gradient-based methods use parametric encodings. We believe that the GP is an instance of the framework that helps to show this connection.
>
> "The interpretation of MAML as a special case of information bottleneck (section 2.2) is not particularly novel: as the paper mentions in section 4, any previous works (LLAMA, probabilistic MAML, bayesian MAML, iMAML)...."
>
> Thanks for this comment. Yes we agree that there exists different ways to obtain MAML.  However, we do claim that the information bottleneck view of MAML, as explained in Section 2.2, is an interesting and simple way to view  MAML.  Notice also the information
> bottleneck provides a rather simple way to understand/interpret  transductive and non-transductive settings (an issue often causing confusion in the literature) as we discuss in Appendix B.
>
> "The experimental results are mixed, and the paper doesn't give much of an interpretation of these results... "
>
> We have updated the experiments to add more results regarding the data efficiency aspect of GP method versus MAML. We also added Table 2 that shows the proposed GP method can give better calibrated probabilities as measured  by negative log-likelihood scores. From Fig2 our conclusion is that regarding data efficiency the GP shows a similar ability for both regression and classification problems.
>
> "Given the relation to other probabilistic extensions of MAML, perhaps comparing against at least one of them would have made experiments more informative."
>
> We will try to reproduce and re-run one of these methods based on available code. We prefer not to quote results from other papers unless we are certain that the neural architecture (including how batch-norm was used etc) and experimental protocol  was exactly the same as Finn et al that is currently the basis for all comparisons.
>
> "The experiments about the efficiency regarding the amount of data (Figure 2 right) are promising, but this claim would be more significant if evaluated on multiple tasks."
>
> We  added mini-Imagenet in this plot and more discussion in the experiments; see last paragraph in this section called "Data efficiency in meta-testing of gradient-based vs GP-based meta learning". A message we wish to convey is that a purely memory-based system can be more data efficient than a gradient-based system.

---

### Official Review · AnonReviewer3 · 2020-11-02
**Interesting but not sure of advantages of proposed method vs. earlier methods**

**Rating:** 4
**Confidence:** 3

**Review:**

The paper presents a method for Bayesian meta-learning. This method combines a NN feature extractor with a Gaussian Process on top. The GP kernel is linear. The information bottleneck is used to motivate a choice of approximate posterior. Using MAML to adapt the NN feature extractor weights improves performance of the composite model for few-shot learning.

---
Things I liked:
- This paper was interesting and on an important topic.
- The information bottleneck seems some to fix some issues with variational inference takes on Bayesian meta learning (e.g. Amortized Bayesian Meta-Learning, Ravi & Beatson, 2019). The VI approach in that paper doesn't distinguish train/support and validation/query sets, and doesn't motivate the weight beta which is placed on the prior, while the information bottleneck does.
- The particular information bottleneck motivated posterior used with the GP is not specific to classification, regression, or other particular structure of prediction problem.
- I liked that the paper had plots of performance vs number of data points, as opposed to only reporting error on the full set of training tasks
- The proposed method leads to a bump in accuracy over MAML for augmented omniglot

---
There are potential areas of concern which I would like the authors to respond to. If the concerns are sufficiently addressed (or turn out to be due to my misunderstanding), I could be willing to significantly increase my score.

Main concerns:
1. What is the objective of the proposed method? Is it to get better test error than MAML, or to quantify uncertainty, or something else? If the point is to get better test error, it seems unfair to compare only to vanilla MAML given the considerable amount of literature and number of improvements in the years since. To build a case for an advantage in terms of accuracy, the paper should compare to a wider range of stronger/more recent baselines (e.g. "Bayesian Few-Shot Classification with One-vs-Each
Pólya-Gamma Augmented Gaussian Processes", Snell & Zemel 2020, does a good job of this).
2. If the motivation for the current proposed method is more than just improving the test error on these benchmarks, this motivation should be clearly explained and/or measured. The method proposed in the paper adds a lot of complexity on top of MAML and variants, so I think the paper needs to build a strong case and clarify exactly which settings this method is preferable for.
3.  One paper the authors could compare to and cite is "Meta-learning with differentiable closed-form solvers", Bertinetto et al 2018, which -- as with the current paper -- combines NN features with a closed-form linear model, and which significantly outperforms MAML.
4. The authors could also compare to the other papers they cite in Section 4, e.g. Snell & Zemel 2020 significantly outperform all the baselines and the proposed method from the current paper's results.
5. Can the authors discuss the lack of data augmentation used for Augmented Omniglot for the GP methods? It strikes me that the data augmentation is likely part of what makes Augmented Omniglot hard. In this case, removing the data augmentation for the GP methods would be giving the GP methods an unfair advantage, which could explain almost all of the GP methods' superiority on this task. A fair comparison should also compare to MAML(+variants) without data aug, and/or GP methods with data aug
6. Can the authors comment on the choice of a linear kernel for the GP? This surprised me, as for regression using a GP with a linear kernel is the same thing as doing Bayesian linear regression. Bayesian linear regression can be done exactly in O(N) time instead of O(N^3) for linear-kernel GP, so it seems strange not to use the LR formulation. It also seems surprising to have to jump through hoops to use an information bottleneck-motivated posterior in a case where we could do exact Bayesian inference cheaply. Of course, in a classification setting one needs to use approximations like Polya-Gamma augmentation, variational inference, or similar.
7. Why only show accuracy vs data points (fig 2 right) for GP not GP-MAML, and why only for augmented omniglot? This is an interesting plot which I would like to see more of in few-shot learning papers.

---
Other points of feedback:
1. I think that whenever the GP method is referred to (title, abstract, text, results tables, plots...), it should be made clear that this is a composition of a neural network feature extractor and a GP. Yes, the NN feature extractor can be seen as part of a very particular choice of kernel for the GP, and this is explained in Section 3. However, someone skimming the paper might not realize the existence of this NN at all, which would make the results very surprising. I think the paper should endeavor to give readers the correct impression on a skim.
2. The set up in Section 2 (and "Stochastic MAML") is extremely similar to Amortized Bayesian Meta-Learning, Ravi & Beatson, 2019. As mentioned, the current submission fixes some issues with that paper, but due the similarity it wouldn't hurt to cite them.
3. The paper should explain the Augmented Omniglot benchmark better in the text (or at least refer the reader to the appendix, where it is explained). Flennerhag et al who introduced the benchmark did not use the term "Augmented Omniglot", so simply citing that paper when the term is mentioned is insufficient and left me confused before I found the discussion in the appendix, although I do think this is a good name for the benchmark.
4. How was the NN+GP method trained? I imagine this was done by backpropogating through the GP fitting procedure to get the gradient of the query-set error for training tasks (or of the IB objective for training tasks) and using this to optimize the NN weights. It would be nice to have this spelled out, as well as how this gradient was computed (whether it is simple or required some non-trivial tricks).

---

> ### Author Response · Authors · 2020-11-18
> **response to the main comments**
>
> We would like to response to the main comments (1 to 7)
>
> 1.  The main objective of our paper is to derive an information bottleneck principle for meta learning. We believe this is a valid methodological contribution. Then to justify this as a valid principle we have fully applied it to GPs. Our method does not provide SOTA classification results (although we are not aware of any published work reporting better accuracy in Augmented Omniglot than our GP method; see Table 8 in Appendix for results with the more advanced architecture). We believe that a clear message from our experiments is that our GP method can be significantly more data efficient than gradient-based methods such as MAML, with better calibrated probabilities.  We have updated the paper to add evidence that supports this claim.  We would be happy to compare with Snell & Zemel 2020. Although given that this work is co-current with ours,  we are not aware if there is code available to be able to re-run this method and include it our ablation study.  We can speculate that Snell & Zemel 2020 provides better classification accuracy due to this really efficient One-vs-Each Pólya-Gamma Augmentation, which  however is very specific to classification.
>
> 2. As mentioned above we have updated the paper to emphasise the data efficiency aspect of our method and also its ability to provide  better predictive uncertainties.
>
> 3. Regarding Bertinetto et al 2018 work their results are not comparable to our results in Table 1 because they use a different architecture than  Finn et al (2017),  that includes Dropout, different convolution layer sizes and leaky-Relu; see Section 4.2 in  Bertinetto et al 2018. Our results are based exactly on the same architecture and experimental setup in  Finn et al (2017). In Table 9 in the Appendix we report significant better results with more advanced architecture (which obviously for fair comparison should be compared only with methods using the same architecture and experimental setup).  We will  add a reference to Bertinetto et al 2018 regarding the similarly of the linear deep kernel.
>
>  4. We could quote classification accuracies from other papers as suggested by the reviewer. But does this add any useful information for our paper? Notice that for MAML we have fully reproduced the MAML results and get for example NNL scores on top of accuracies.
>
> 5. Regarding Augmented Omniglot for meta-training we used only the 300 task support points without further data augmentation. This proves to be sufficient for meta-training of the deep kernel feature vector, without data augmentation. Given that a GP is memory-based system then if we use data augmentation on top of the initial 300 task support points, then GP will need to memorise more data points, which while could improve further performance it could make meta-training more expensive.  However, for meta testing GP uses data augmentation exactly as MAML and both methods process exactly the same amount of data  as shown in Fig2. We also want to point out that MAML without data augmentation in meta-training will work much worse than the results reported.
>
> 6. We prefer the linear kernel since it makes the GP as close as possible to a non-GP system that uses the same DNN feature vector. The linear kernel  it is also computationally convenient. However, while exact Bayesian linear regression can be done linearly wrt the data points N, it has O(M^3) complexity over the size of feature vector so it is often much slower than O(N^3) in few-shot learning. For example, in mini-Imagenet N=5,K=1 the feature vector has size M=801, while N=5, so it is clearly preferable to decompose (with Cholesky) a 5 x 5 matrix than a 801 x 801 one. The linear kernel allows to choose between O(M^3) and O(N^3) based on whatever is faster at each occasion. A nice discussion that we find useful on this issue is from "Prediction with Gaussian Processes: From Linear Regression to Linear Prediction and Beyond C. K. I. Williams Aston University, UK. In "Learning and Inference in Graphical Models", ed. M. I. Jordan, Kluwer, 1998."
>
> 7. We have added mini-Imagenet in this plot and more discussion in the experiments. The message we wish to convey is that a purely memory-based system can be more data efficient than a gradient-based system, thus we prefer to keep only GP in this plot. GP+MAML can be better than GP, but at the same time since GP+MAML is also a gradient-based system its behaviour will depend on the learning rate and the inner loop size.   Fig4 in the Appendix shows how GP+MAML  compares with GP as a function of the inner loop size of GP+MAML on mini-Imagenet.
>
> Also to clarify 4) from "Other points of feedback:" Meta-training is done  in a completely end-to-end fashion maximising VIB over the neural representation $\phi(x;\theta)$ using automatic differentiation. Appendix C fully describes this. In particular, Appendix C.4 gives the full objective for few-shot classification.

---

### Decision · Program_Chairs · 2021-01-07
**Final Decision**

**Decision:**

Reject

**Comment:**

Information bottleneck is a well-known principle that is used for clustering, dimensionality reduction, and recently deep learning. It finds a compressed representation of input X while retaining most information on the response Y. This paper addresses an attempt to interpret the meta-learning using the information bottleneck. In addition, a GP-based meta-learning method is also proposed.
The topic itself is interesting without any doubt. However, most of reviewers have serious concerns about this work, which is summarized below. First of all, two components of this paper (IB and GP-based meta-learning) do not provide a coherent message.  While the IB interpretation is emphasized in the beginning of this paper, the main point seems to that GP-based methods can be more data efficient than gradient-based meta learning. There does not much point to GP+MAML or IB interpretation of MAML.  Experiments are not strong enough, although a few ones are added during the author responses. During the discussion with reviewers, no one support this work, so I do not have choice but to suggest rejection.